# UNDERSTANDING AND PREVENTING CAPACITY LOSS IN REINFORCEMENT LEARNING

**Clare Lyle**
Department of Computer Science
University of Oxford*

**Mark Rowland & Will Dabney**
DeepMind

## ABSTRACT

The reinforcement learning (RL) problem is rife with sources of non-stationarity, making it a notoriously difficult problem domain for the application of neural networks. We identify a mechanism by which non-stationary prediction targets can prevent learning progress in deep RL agents: *capacity loss*, whereby networks trained on a sequence of target values lose their ability to quickly update their predictions over time. We demonstrate that capacity loss occurs in a range of RL agents and environments, and is particularly damaging to performance in sparse-reward tasks. We then present a simple regularizer, Initial Feature Regularization (InFeR), that mitigates this phenomenon by regressing a subspace of features towards its value at initialization, leading to significant performance improvements in sparse-reward environments such as Montezuma's Revenge. We conclude that preventing capacity loss is crucial to enable agents to maximally benefit from the learning signals they obtain throughout training.

## 1 INTRODUCTION

Deep reinforcement learning has achieved remarkable successes in a variety of tasks (Mnih et al., 2015; Moravčík et al., 2017; Silver et al., 2017; Abreu et al., 2019), but its impressive performance is mirrored by its brittleness and sensitivity to seemingly innocuous design choices (Henderson et al., 2018). In sparse-reward environments in particular, even different random seeds of the same algorithm can attain dramatically different performance outcomes. This presents a stark contrast to supervised learning, where existing approaches are reasonably robust to small hyperparameter changes, random seed inputs, and GPU parallelisation libraries. Much of the brittleness of deep RL algorithms has been attributed to the non-stationary nature of the prediction problems to which deep neural networks are applied in RL tasks. Indeed, naive applications of supervised learning methods to the RL problem may require explicit correction for non-stationarity and bootstrapping in order to yield similar improvements (Bengio et al., 2020; Raileanu et al., 2020).

We hypothesize that the non-stationary prediction problems agents face in RL may be a driving force in the challenges described above. RL agents must solve a sequence of similar prediction tasks as they iteratively improve their value function accuracy and their policy (Dabney et al., 2021). Solving each subproblem (at least to the extent that the agent's policy is improved) in this sequence is necessary to progress to the next subproblem. Ideally, features learned to solve one subproblem would enable forward transfer to future problems. However, prior work on both supervised and reinforcement learning (Ash & Adams, 2020; Igl et al., 2021; Fedus et al., 2020) suggests that the opposite is true: networks trained on a sequence of similar tasks are prone to overfitting, exhibiting *negative* transfer.

The principal thesis of this paper is that over the course of training, deep RL agents lose some of their capacity to quickly fit new prediction tasks, and in extreme cases this capacity loss prevents the agent entirely from making learning progress. We present a rigorous empirical analysis of this

---

*Correspondence to `clare.lyle@cs.ox.ac.uk`

phenomenon which considers both the ability of networks to learn new target functions via gradient-based optimization methods, and their ability to linearly disentangle states' feature representations. We confirm that agents' ability to fit new target functions declines over the course of training in several environments from the Atari suite (Bellemare et al., 2013) and non-stationary reward prediction tasks. We further find that the ability of representations to linearly distinguish different states, a proxy for their ability to represent certain functions, quickly diminishes in sparse-reward environments, leading to representation collapse, where the feature outputs for every state in the environment inhabit a low-dimensional – or possibly even zero – subspace. Crucially, we find evidence that sufficient capacity is a *necessary* condition in order for agents to make learning progress. Finally, we propose a simple regularization technique, Initial Feature Regularization (InFeR), to prevent representation collapse by regressing a set of auxiliary outputs towards their value under the network's initial parameters. We show that this regularization scheme mitigates capacity loss in a number of settings, and also enables significant performance improvements in a number of RL tasks.

One striking take-away from our results is that agents trained on so-called 'hard exploration' games such as Montezuma's Revenge can attain significant improvements over existing competitive baselines *without* using smart exploration algorithms, given a suitable representation learning objective. This suggests that the poor performance of deep RL agents in sparse-reward environments is not solely due to inadequate exploration, but rather also in part due to poor representation learning. Investigation into the interplay between representation learning and exploration, particularly in sparse-reward settings, thus presents a particularly promising direction for future work.

## 2    BACKGROUND

We consider the reinforcement learning problem wherein an agent interacts with an environment formalized by a Markov Decision Process $\mathcal{M} = (\mathcal{X}, \mathcal{A}, R, \mathcal{P}, \gamma)$, where $\mathcal{X}$ denotes the state space, $\mathcal{A}$ the action space, $R$ the reward function, $\mathcal{P}$ the transition probability function, and $\gamma$ the discount factor. We will be primarily interested in *value-based* RL, where the objective is to learn the value function $Q^\pi : \mathcal{X} \times \mathcal{A} \to \mathbb{R}$ associated with some (possibly stochastic) policy $\pi : \mathcal{X} \to \mathscr{P}(\mathcal{A})$, defined as $Q^\pi(x, a) = \mathbb{E}_{\pi, \mathcal{P}}[\sum_{k=0}^\infty \gamma^k R(x_k, a_k) | x_0 = x, a_0 = a]$. In particular, we are interested in learning the value function associated with the optimal policy $\pi^*$ which maximizes the expected discounted sum of rewards from any state.

In Q-Learning (Watkins & Dayan, 1992), the agent performs updates to minimize the distance between a predicted action-value function $Q$ and the bootstrap target defined as

$$\mathcal{T}Q(x, a) = \mathbb{E}[R(x_0, a_0) + \gamma \max_{a' \in \mathcal{A}} Q(x_1, a') | x_0 = x, a_0 = a] . \tag{1}$$

In most practical settings, updates are performed with respect to sampled transitions rather than on the entire state space. The target can be computed for a sampled transition $(x_t, a_t, r_t, x_{t+1})$ as $\hat{\mathcal{T}}Q(x_t, a_t) = r_t + \gamma \max_a Q(x_{t+1}, a)$.

When a deep neural network is used as a function approximator (the deep RL setting), $Q$ is defined to be the output of a neural network with parameters $\theta$, and updates are performed by gradient descent on sampled transitions $\tau = (x_t, a_t, r_t, x_{t+1})$. A number of tricks are often used to improve stability: the sample-based objective is minimized following stochastic gradient descent based on minibatches sampled from a replay buffer of stored transitions, and a separate set of parameters $\bar{\theta}$ is used to compute the targets $Q_{\bar{\theta}}(x_{t+1}, a_{t+1})$ which is typically updated more slowly than the network's online parameters. This yields the following loss function, given a sampled transition $\tau$:

$$\ell_{TD}(Q_\theta, \tau) = (R_{t+1} + \gamma \max_{a'} Q_{\bar{\theta}}(X_{t+1}, a') - Q_\theta(X_t, A_t))^2 . \tag{2}$$

In this work we will be interested in how common variations on this basic learning objective shape agents' learning dynamics, in particular the dynamics of the learned *representation*, or *features*. We will refer to the outputs of the final hidden layer of the network (i.e. the penultimate layer)

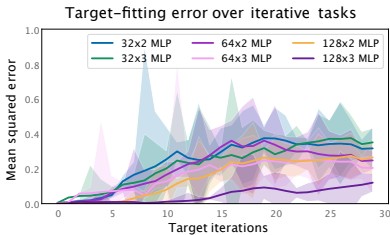

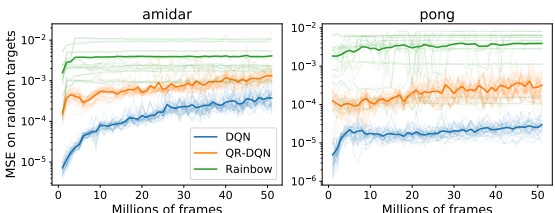

Figure 1: Networks trained to fit a sequence of different targets on MNIST data see increasing error on new target functions with the number of tasks.

Figure 2: Networks see reduced ability to fit new targets over the course of training in two demonstrative Atari environments.

as its features, denoted $\phi_\theta(x)$. Our choice of the penultimate layer is motivated by prior literature studying representations in RL (Ghosh & Bellemare, 2020; Kumar et al., 2021), although many works studying representation learning consider the outputs of earlier layers as well. In general, the features of a neural network are defined to be the outputs of whatever layer is used to compute additional representation learning objectives.

## 3 CAPACITY LOSS

Each time a value-based RL agent discovers a new source of reward in its environment or, in the case of temporal difference methods, updates its value estimate, the prediction problem it needs to solve changes. Over the course of learning, such an agent must solve a long sequence of target prediction problems as its value function and policy evolve. Studies of neural networks in supervised learning suggest that this sequential fitting of new targets may be harmful to a network's ability to adapt to new targets (Achille et al., 2018, see Section 5 for further details, e.g.). This presents a significant challenge to deep RL agents undergoing policy improvement, for which it is necessary to quickly make significant changes to the network's predictions even late in the training process. In this section, we show that training on a sequence of prediction targets *can* lead to a reduced ability to fit new targets in deep neural networks, a phenomenon that we term *capacity loss*. Further, we show that an agent's inability to quickly update its value function to distinguish states presents a barrier to performance improvement in deep RL agents.

### 3.1 TARGET-FITTING CAPACITY

The parameters of a neural network not only determine the network's current outputs, but also influence how these outputs will evolve over time. A network which outputs zero because its final-layer weights are zero will evolve differently from one whose ReLU units are fully saturated at zero despite both outputting the same function Maas et al. (2013) – in particular, it will have a much easier time adapting to new targets. It is this capacity to fit new targets that is crucial for RL agents to obtain performance improvements, and which frames our perspective on representation learning. We are interested in identifying when an agent's current parameters are flexible enough to allow it to perform gradient updates that meaningfully change its predictions based on new reward information in the environment or evolving bootstrap targets, a notion formalized in the following definition.

**Definition 1** (Target-fitting capacity). *Let $P_X \in \mathscr{P}(X)$ be some distribution over inputs $X$ and $P_\mathcal{F}$ a distribution over a family of real-valued functions $\mathcal{F}$ with domain $X$. Let $\mathcal{N} = (g_\theta, \theta_0)$ represent the pairing of a neural network architecture with some initial parameters $\theta_0$, and $\mathcal{O}$ correspond to an optimization algorithm for supervised learning. We measure the* target-fitting capacity *of $\mathcal{N}$ under the optimizer $\mathcal{O}$ to fit the data-generating distribution $\mathcal{D} = (P_X, P_\mathcal{F})$ as follows:*

$$\mathcal{C}(\mathcal{N}, \mathcal{O}, \mathcal{D}) = \mathbb{E}_{f \sim P_\mathcal{F}}[\mathbb{E}_{x \sim P_X}[(g_{\theta'}(x) - f(x))^2]] \quad \text{where } \theta' = \mathcal{O}(\theta_0, P_X, f). \quad (3)$$

Our definition of capacity measures the ability of a network to reach a new set of targets within a limited optimization budget from its current parameters and optimizer state. The choice of optimization budget and target distribution are left as hyperparameters, and different choices result in different notions of capacity. In reinforcement learning we ultimately care about the network's ability to fit its Bellman targets quickly, however the ability on its own will not necessarily be a useful measure: for example, a network which can only output the zero function will attain low Bellman error immediately on a sparse-reward environment, but will fail to produce useful updates to improve the policy. Our evaluations of this measure will use target functions that are independent of the current network parameters to avoid these pathologies; the effect of this choice is explored further in Appendix B.2.

The process of training a neural network to fit a set of labels must by necessity change some properties of the network. Works studying the information bottleneck principle (Tishby & Zaslavsky, 2015), for example, identify a compression effect of training on the latent representation, where inputs with similar labels are mapped to similar feature vectors. This compression can benefit generalization on the current task, but in the face of the rapidly-changing nature of the targets used in value iteration algorithms may harm the learning process by impeding the network's ability to fit new targets. This motivates two hypotheses. First: that networks trained to iteratively fit a sequence of dissimilar targets will lose their capacity to fit new target functions (**Hypothesis 1**), and second: the non-stationary prediction problems in deep RL also result in capacity loss (**Hypothesis 2**).

To evaluate **Hypothesis 1**, we construct a series of toy iterative prediction problems on the MNIST data set, a widely-used computer vision benchmark which consists of images of handwritten digits and corresponding labels. We first fit a series of labels computed by a randomly initialized neural network $f_\theta$: we transform input-label pairs $(x, y)$ from the canonical MNIST dataset to $(x, f_\theta(x))$, where $f_\theta(x)$ is the network output. To generate a new task, we simply reinitialize the network. Given a target function, we then train the network for a fixed budget from the parameters obtained at the end of the previous iteration, and repeat this procedure of target initialization and training 30 times. We use a subset of MNIST inputs of size 1000 to reduce computational cost. In Figure 1 we see that the networks trained on this task exhibit decreasing ability to fit later target functions under a fixed optimization budget. This effect is strongest in the smaller networks, matching the intuition that solving tasks which are more challenging for the network will result in greater capacity loss. We consider two other tasks in Appendix B.2, obtaining similar results, as well as a wider range of architectures. We find that sufficiently over-parameterized networks (on the order of one million parameters for a task with one thousand data points) exhibit positive forward transfer, however models which are not over-parameterized relative to the task difficulty consistently exhibit increasing error as the number of targets trained on grows. This raises a question concerning our second hypothesis: are the deep neural networks used by value-based RL agents on popular benchmarks in the over- or under-parameterized regime?

To evaluate **Hypothesis 2**, we train an agent's network checkpoints sampled over the course of training to fit randomly generated target functions. We provide full details of this procedure in Appendix C. We generate target functions by randomly initializing neural networks with new parameters, and use the outputs of these networks as targets for regression. We then load initial parameters from an agent checkpoint at some time $t$, sample inputs from the replay buffer, and regress on the random target function evaluated on these inputs. We then evaluate the mean squared error after training for fifty thousand steps. We consider a DQN (Mnih et al., 2015), a QR-DQN (Dabney et al., 2018), and a Rainbow agent (Hessel et al., 2018). We observe in all three cases that as training progresses agents' checkpoints on average get modestly worse at fitting these random targets in most environments; due to space limitations we only show two representative environments where this phenomenon occurs in Figure 2, and defer the full evaluation to Appendix C.3.

## 3.2    REPRESENTATION COLLAPSE AND PERFORMANCE

The notion of capacity in Definition 1 measures the ability of a network to *eventually* represent a given target function. This definition reflects the intuition that capacity should not increase over time.

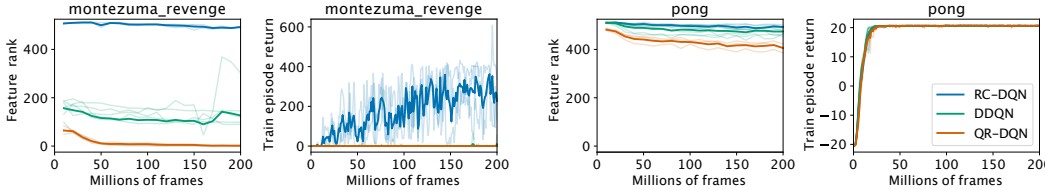

Figure 3: Feature rank and performance over the course of training for Montezuma's Revenge (left) and Pong (right). We observe that feature rank is higher for environments and auxiliary tasks which provide denser reward signals than for sparse reward problems.

However, deep RL agents must *quickly* update their predictions in order to make efficient learning progress. We present an alternate measure of capacity that captures this property which we call the feature rank, as it corresponds to an approximation of the rank of a feature embedding. Intuitively, the feature rank measures how easily states can be distinguished by updating only the final layer of the network. This approximately captures a network's ability to quickly adapt to changes in the target function, while being significantly cheaper to estimate than Definition 1.

**Definition 2** (Feature rank). *Let $\phi : X \to \mathbb{R}^d$ be a feature mapping. Let $\mathbf{X}_n \subset X$ be a set of $n$ states in $X$ sampled from some fixed distribution $P$. Fix $\varepsilon \geq 0$, and let $\phi(\mathbf{X}_n) \in \mathbb{R}^{n \times d}$ denote the matrix whose rows are the feature embeddings of states $x \in \mathbf{X}_n$. Let $\mathrm{SVD}(M)$ denote the multiset of singular values of a matrix $M$. The feature rank of $\phi$ given input distribution $P$ is defined to be*

$$\rho(\phi, P, \epsilon) = \lim_{n \to \infty} \mathbb{E}_{\mathbf{X}_n \sim P}[|\{\sigma \in \mathrm{SVD}\left(\frac{1}{\sqrt{n}}\phi(\mathbf{X}_n)\right)|\sigma > \varepsilon\}|] \tag{4}$$

*for which a consistent estimator can be constructed as follows, letting $\mathbf{X} \subseteq X, |\mathbf{X}| = n$*

$$\hat{\rho}_n(\phi, \mathbf{X}, \epsilon) = |\{\sigma \in \mathrm{SVD}\left(\frac{1}{\sqrt{n}}\phi(\mathbf{X})\right)|\sigma > \varepsilon\}|. \tag{5}$$

The numerical feature rank (henceforth abbreviated to feature rank) is equal to the dimension of the subspace spanned by the features when $\varepsilon = 0$ and the state space $\mathcal{X}$ is finite, and its estimator is equal to the numerical rank (Golub et al., 1976; Meier & Nakatsukasa, 2021) of the sampled feature matrix. For $\epsilon > 0$, it throws away small components of the feature matrix. We show that $\rho$ is well-defined and that $\hat{\rho}_n$ is a consistent estimator in Appendix A.1. Our analysis of the feature rank resembles that of Kumar et al. (2021), but differs in two important ways: first, our estimator does not normalize by the maximal singular value. This allows us to more cleanly capture *representation collapse*, where the network features, and thus also their singular values, converge to zero. Second, we are interested in the capacity of agents with unlimited opportunity to interact with the environment, rather than in the data-limited regime. We compare our findings on feature rank against the *srank* used in prior work in Appendix B.2.

In our empirical evaluations, we train a double DQN (DDQN) agent, a quantile regression (QRDQN) agent, and a double DQN agent with an auxiliary random cumulant prediction task (RC DQN) (Dabney et al., 2021), on environments from the Atari suite, then evaluate $\hat{\rho}_n$ with $n = 5000$ on agent checkpoints obtained during training. We consider two illustrative environments: Montezuma's Revenge (sparse reward), and Pong (dense reward), deferring two additional environments to Appendix C.3. We run 3 random seeds on each environment-agent combination.

We visualize agents' feature rank and performance in Figure 3. Non-trivial prediction tasks, either value prediction in the presence of environment rewards or auxiliary tasks, lead to higher feature rank. In Montezuma's Revenge, the higher feature rank induced by RC DQN corresponds to higher performance, but this auxiliary loss can have a detrimental effect on learning progress in complex, dense-reward games presumably due to interference between the random rewards and the true learning objective. Unlike in target-fitting capacity, we only see a consistent downward trend in sparse-reward environments, where a number of agents, most dramatically QR-DQN, exhibit representation collapse. We discuss potential mechanisms behind this trend in Appendix A.2.

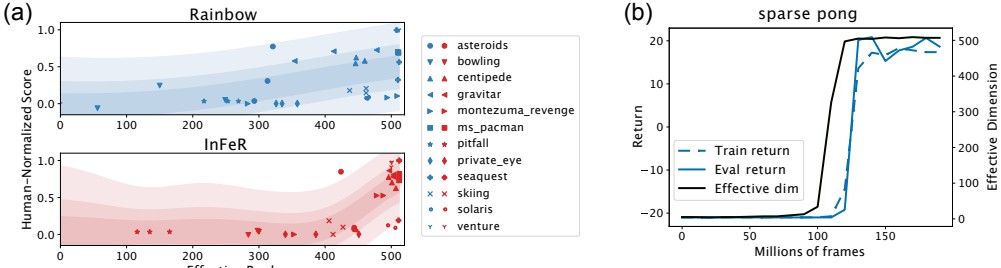

Figure 4: **(a)**: Agent capacity vs human-normalized score in games where Rainbow does not achieve superhuman performance. While feature rank does not appear to solely determine agent performance, there is a positive correlation between feature rank and human-normalized score. Bottom row contains Rainbow agents trained with the regularizer presented in Equation 6. **(b)** An 'unlucky' seed from our evaluations on the sparsified version of Pong, where learning progress occurs only after the agent recovers from representation collapse.

Figure 4a reveals a correlation between learning progress and feature rank for the Rainbow agent (Hessel et al., 2018) trained on challenging games in the Atari 2600 suite where it fails to achieve human-level performance; this trend is also reflected for other agents described in the next section. The points on the scatterplot largely fall into two clusters: those with low feature rank, which attain less than half of the average human score, and those with high feature rank, which tend to attain higher scores. Having a sufficiently high feature rank thus appears to be a *necessary* condition for learning progress, as demonstrated by the learning curves shown in Figure 4b, which highlights an unlucky agent trained on a variant of Pong (described in Appendix C.2) which experienced representation collapse, and only solved the task *after* it had overcome this collapse. However, high feature rank does not appear to be sufficient for learning progress. Other properties of an agent, such as its ability to perform accurate credit assignment, the stability of its update rule, the suitability of its optimizer, its exploration policy, and countless others, must be appropriately tuned to a given task in order for progress to occur. Simply mapping inputs to a relatively uniform distribution in feature space will not overcome failures in other components of the RL problem. An agent must be able to both collect useful learning signals from the environment and effectively update its predictions in response to those signals in order to make learning progress. This section has shown that at least in some instances poor performance can be attributed to the latter property.

## 4 INFER: MITIGATING CAPACITY LOSS WITH FEATURE REGULARIZATION

The previous section showed that capacity loss occurs in deep RL agents trained with online data, and in some cases appears to be a bottleneck to performance. We now consider how it might be mitigated, and whether explicitly regularizing the network to preserve its initial capacity improves performance in environments where representation collapse occurs. Our approach involves a function-space perspective on regularization, encouraging networks to preserve their ability to output linear functions of their features at initialization.

### 4.1 INFER: FEATURE-SPACE REGULARIZATION

Much like parameter regularization schemes seek to keep *parameters* close to their initial values, we wish to keep a network's ability to fit new targets close to its initial value. We motivate our approach with the intuition that a network which has preserved the ability to output functions it could easily fit at initialization should be better able to adapt to new targets. To this end, we will regress a set of network *outputs* towards the values they took at initialization. Our method, Initial Feature Regularization (InFeR), applies an $\ell_2$ regularization penalty on the output-space level by regressing

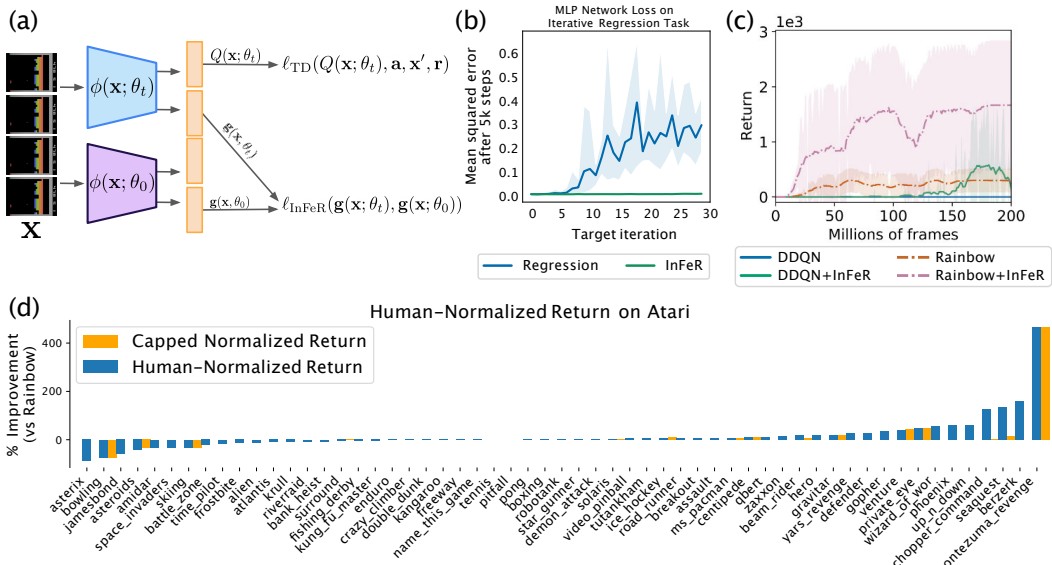

Figure 5: **(a)** Visualization of InFeR. **(b)** Analysis of the effect of InFeR on capacity loss. **(c)** Effect of InFeR on performance in Montezuma's Revenge with respect to Rainbow and Double DQN baselines. **(d)** Performance of InFeR relative to Rainbow on all 57 Atari games.

a set of auxiliary network output heads to match their values at initialization. Similar perspectives have been used to prevent catastrophic forgetting in continual learning (Benjamin et al., 2019).

In our approach, illustrated in Figure 5, we begin with a fixed deep Q-learning neural network with parameters $\theta$, and modify the network architecture by adding $k$ auxiliary linear prediction heads $g_i$ on top of the feature representation $\phi_\theta$. We take a snapshot of the agent's parameters at initialization $\theta_0$, and use the outputs of the $k$ auxiliary heads under these parameters as auxiliary prediction targets. We then compute the mean squared error between the outputs of the heads under the current parameters $g_i(x; \theta_t)$ and their outputs at initialization $g_i(x; \theta_0)$. This approach has the interpretation of amplifying and preserving subspaces of the features that were present at initialization. In practice, we find that scaling the auxiliary head outputs by a constant $\beta$ increases this amplification effect. This results in the following form of our regularization objective, where we let $\mathcal{B}$ denote the replay buffer sampling scheme used by the agent:

$$\mathcal{L}_{\text{InFeR}}(\theta, \theta_0; \mathcal{B}, \beta) = \mathbb{E}_{x \sim \mathcal{B}}\left[\sum_{i=1}^{k}(g_i(x; \theta) - \beta g_i(x; \theta_0))^2\right]. \quad (6)$$

We evaluate the effect of incorporating this loss in both DDQN (Van Hasselt et al., 2016) and Rainbow (Hessel et al., 2018) agents, and include the relative performance improvement obtained by the InFeR agents over Rainbow on 57 games from the Atari 2600 suite in Figure 5, deferring the comparison to DDQN, where the regularizer improved performance slightly on average but only yielded significant improvements on sparse-reward games, to the appendix. We observe a net improvement over the Rainbow baseline by incorporating the InFeR objective, with significant improvements in games where agents struggle to obtain human performance. The evaluations in Figure 5 are for $k = 10$ heads with $\beta = 100$ and $\alpha = 0.1$, and we show the method's robustness to these hyperparameters in Appendix C.1. We further observe in Figure 5 that the InFeR loss reduces target-fitting error on the non-stationary MNIST prediction task described in the previous section. We show in Appendix C.2 that InFeR tends to increase the feature rank of agents trained on the Atari domain over the entire course of training; we study the early training period in Appendix C.3.

The striking improvement obtained in the sparse-reward Montezuma's Revenge environment begs the question of whether such results can be replicated in other RL agents. We follow the same experimental procedure as before, but now use the DDQN agent; see Figure 5. We find that adding InFeR to the DDQN objective produces a similar improvement as does adding it to Rainbow, leading the DDQN agent, which only follows an extremely naive $\epsilon$-greedy exploration strategy and obtains zero reward at all points in training, to exceed the performance of the noisy networks approach taken by Rainbow in the last 40 million training frames. This leads to two intriguing conclusions: first, that agents which are explicitly regularized to prevent representation collapse *can* make progress in sparse reward problems without the help of good exploration strategies; and second, that this form of regularization yields significantly larger performance improvements in the presence of additional algorithm design choices that are designed to speed up learning progress.

## 4.2 Understanding How InFeR Works

While InFeR improves performance *on average* across the Atari games, its improvements are concentrated principally on games where the baseline rainbow agent performs significantly below the human baseline. It further slows down progress in a subset of environments such as *Asteroids* and *Jamesbond*. We now investigate two hypothesized mechanisms by which this regularizer may shape the agent's representation, in the hopes of explaining this differential effect on performance. **Hypothesis 1:** InFeR improves performance by preserving a random subspace of the representation that the final linear layer can use to better predict the value function. The effect of the regularizer on other aspects of the representation learning dynamics does not influence performance. **Hypothesis 2:** The InFeR loss slows down the rate at which the learned features at every layer of the network can drift from their initialization in function space, improving the learning dynamics of the entire network to prevent feature collapse and over-fitting to past targets. The precise subspace spanned by the auxiliary weights is not directly useful to value function estimation.

To evaluate Hypothesis 1, we concatenate the outputs of a randomly initialized network to the feature outputs of the network used to learn the Q-function, and train a linear layer on top of these joint learned and random features. If Hypothesis 1 were true, then we would expect this architecture to perform comparably to the InFeR agents, as the final linear layer has access to a randomly initialized feature subspace. Instead, Figure 6 shows that the performance of the agents with access to the random features to be comparable to that of the vanilla Rainbow agents, confirming that the effect of InFeR on earlier layers is crucial to its success.

We now consider Hypothesis 2. InFeR limits the degrees of freedom with which a network can collapse its representation, which may reduce the flexibility of the network to make the changes necessary to fit new value functions, slowing down progress in environments where representation collapse is not a concern. In such cases, increasing the dimension of the layer to which we apply InFeR should give the network more degrees of freedom to fit its targets, and so reduce the performance gap induced by the regularization. We test this hypothesis by doubling the width of the penultimate network layer and comparing the performance of InFeR and Rainbow on games where

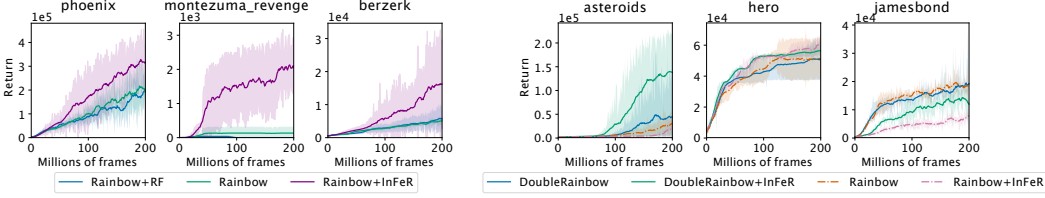

Figure 6: Left: agent performance does not improve over baseline when random features are added to the representation. Right: doubling the width of the neural network narrows the performance gap in games on which InFeR under-performed relative to Rainbow.

InFeR hurt performance in the original network. We refer to this agent as DoubleRainbow. We see in Figure 6 that increasing the network's size reduces, eliminates, or in some cases reverses the performance gap induced by InFeR in the smaller architecture. We therefore conclude that the principal mechanism by which InFeR affects performance is by regularizing the entire network's learning dynamics.

## 5   RELATED WORK

Suitably designed auxiliary tasks have been shown to improve performance and encourage learned representations to satisfy desirable properties in a wide range of settings (Jaderberg et al., 2017; Veeriah et al., 2019; Gelada et al., 2019; Machado et al., 2018), with further insight given by prior analysis of the geometry (Bellemare et al., 2019) and stability (Ghosh & Bellemare, 2020) of value functions in RL. Our analysis of linear algebraic properties of agents' representations is complemented by prior works which leverage similar ideas to analyze implicit under-parameterization (Kumar et al., 2021) and spectral normalization (Gogianu et al., 2021) in deep RL agents, and by the framework proposed by Lyle et al. (2021) to study learning dynamics in deep RL agents. In contrast to prior work, which treats the layers of the network which come before the features as a black box, we explicitly study the properties and learning dynamics of the whole network.

A separate line of work has studied the effect of interference between sub-tasks in both reinforcement learning (Schaul et al., 2019; Teh et al., 2017; Igl et al., 2021) and supervised learning settings (Sharkey & Sharkey, 1995; Ash & Adams, 2020; Beck et al., 2021). Of particular interest has been catastrophic forgetting, with prior work proposing novel training algorithms using regularization (Kirkpatrick et al., 2017; Bengio et al., 2014; Lopez-Paz & Ranzato, 2017) or distillation (Schwarz et al., 2018; Silver & Mercer, 2002; Li & Hoiem, 2017) approaches. Methods which involve re-initializing a new network have seen particular success at reducing interference between tasks in deep reinforcement learning (Igl et al., 2021; Teh et al., 2017; Rusu et al., 2016; Fedus et al., 2020). A closer relative of our approach is that of Benjamin et al. (2019), which also applies a function-space regularization approach, but which involves saving input-output pairs into a memory bank with the goal of mitigating catastrophic forgetting. Unlike prior work, InFeR seeks to maximize performance on *future* tasks, works without task labels, and incurs a minimal, fixed computational cost independent of the number of prediction problems seen during training.

## 6   CONCLUSIONS

This paper has demonstrated a fundamental challenge facing deep RL agents: loss of the capacity to distinguish states and represent new target functions over the course of training. We have shown that this phenomenon is particularly salient in sparse-reward settings, in some cases leading to complete collapse of the representation and preventing the agent from making learning progress. Our analysis revealed a number of nuances to this phenomenon, showing that larger networks trained on rich learning signals are more robust to capacity loss than smaller networks trained to fit sparse targets. To address this challenge, we proposed a regularizer to preserve capacity, yielding improved performance across a number of settings in which deep RL agents have historically struggled to match human performance. Further investigation into this method suggests that it is performing a form of function-space regularization on the neural network, and that settings where it appears the task reduces performance are actually instances of under-parameterization relative to the difficulty of the environment. Particularly notable is the effect of incorporating InFeR in the hard exploration game of Montezuma's Revenge: its success here suggests that effective representation learning can allow agents to learn good policies in sparse-reward environments even under naive exploration strategies. Our findings open up a number of exciting avenues for future work in reinforcement learning and beyond to better understand how to preserve plasticity in non-stationary prediction tasks.

## ACKNOWLEDGEMENTS

Thanks to Georg Ostrovski, Michael Hutchinson, Joost van Amersfoort, Daniel Guo, Diana Borsa, Anna Harutyunyan, Razvan Pascanu, Caglar Gulcehre, Srivatsan Srvinivasan, and Remi Munos for helpful discussions and feedback on early versions of this paper. CL is supported by an Open Philanthropy AI Fellowship.

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

## A  THEORETICAL RESULTS

### A.1  ESTIMATOR CONSISTENCY

We here show that our estimator of the agent's feature rank is consistent. First recall

$$\left(\frac{1}{\sqrt{n}}\Phi_n\right)^{\top}\left(\frac{1}{\sqrt{n}}\Phi_n\right) = \frac{1}{n}\sum_{i=1}^{n}\phi(x_i)\phi(x_i)^{\top}. \tag{7}$$

The following property of the expected value holds

$$\mathbb{E}_{x\sim P}[\phi(x)\phi(x)^{\top}] = \mathbb{E}\left[\frac{1}{n}\sum_{i=1}^{n}\phi(x_i)\phi(x_i)^{\top}\right]. \tag{8}$$

It is then straightforward to apply the strong law of large numbers. To be explicit, we consider an element of $M = \mathbb{E}[\phi\phi^{\top}]$, $M_{ij}$.

$$\mathbb{E}[(\phi(x)\phi(x)^{\top})_{ij}] = M_{ij} = \mathbb{E}[\phi_i(x)\phi_j(x)] \implies \sum_{k=1}^{n}\frac{1}{n}\phi_i(x_k)\phi_j(x_k) \overset{a.s.}{\to} M_{ij}. \tag{9}$$

Since we have convergence for any $M_{ij}$, we get convergence of the resulting matrix to $M$. Because the singular values of $\Phi$ are the eigenvalues of $M$ and the eigenvalues are continuous functions of that matrix, the eigenvalues of $M_n$ converge to those of $M$ almost surely. Then for almost all values of $\epsilon$, the threshold estimator $N(\lambda_1, \ldots, \lambda_k; \epsilon) = |\{\lambda_i > \epsilon\}|$ will converge to $N(\text{spec}(M); \epsilon)$. Specifically, the estimator will be convergent for all values of $\epsilon$ which are not eigenvalues of $M$ itself.

### A.2  FEATURE DYNAMICS

We apply similar analysis to that of Lyle et al. (2021) to better understand the effect of sparse-reward environments on representation collapse. To do so, we consider the setting where $\Phi_t$ are features and $w_t$ a linear function approximator which jointly parameterize a value function $V_t = \langle \Phi_t(x), w_t \rangle$. We will be interested in studying a continuous-time approximation to TD learning, where the discrete-time expected updates

$$\Phi_t \leftarrow \Phi_t + \alpha \nabla_{\Phi} V_t[(\gamma P^{\pi} - I)V_t + R^{\pi}] \tag{10}$$
$$w_t \leftarrow w_t + \beta \nabla_w V_t(\gamma P^{\pi} - I)V_t + R^{\pi}] \tag{11}$$

are translated into a continuous-time flow, described by the following equations.

$$\partial_t \Phi_t = \alpha(\gamma P^{\pi} - I)\Phi_t(w_t w_t^{\top}) + R^{\pi} w_t^{\top} \tag{12}$$
$$\partial_t w_t = \beta \Phi_t^{\top}[(\gamma P^{\pi} - I)\Phi_t w_t + R^{\pi}], \tag{13}$$

where $P^{\pi} \in \mathbb{R}^{\mathcal{X} \times \mathcal{X}}$ is the matrix of state-transition probabilities under $\pi$, and $R^{\pi} \in \mathbb{R}^{\mathcal{X}}$ is the vector of expected rewards.

One of the key take-aways of prior works is that under certain assumptions, a tabular value function following continuous-time TD dynamics will converge to its limiting value $V^{\pi}$ along the principal components of the environment's transition matrix. In the function-approximation case described above, the dynamics of the features $\Phi_t$ are somewhat more complex. However, it turns out that under certain training regimes, we can obtain similar convergence results for the features. We therefore turn our attention to ensemble prediction, where $M$ linear prediction 'heads', each using a separate weight vector $w_t^m$ ($m = 1, \ldots, M$) are all trained to regress on the TD targets using the shared feature representation of the state as input, resulting in the following dynamics.

$$\partial_t \Phi_t^M = \alpha \sum_{m=1}^{M}(R^{\pi} + \gamma P^{\pi}\Phi_t^M w_t^m - \Phi_t^M w_t^m)(w_t^m)^{\top}, \tag{14}$$

$$\partial_t w_t^m = \beta(\Phi_t^M)^{\top}(R^{\pi} + \gamma P^{\pi}\Phi_t^M w_t^m - \Phi_t w_t^m). \tag{15}$$

We now restate the result of Lyle et al. (2021) regarding the behaviour of the representation in the limit of many ensemble heads.

**Theorem 1** (Lyle et al., 2021). *For $M \in \mathbb{N}$, let $(\Phi_t^M)_{t \geq 0}$ be the solution to Equation 14, with each $w_t^m$ for $m = 1, \ldots, M$ initialised independently from $N(0, \sigma_M^2)$, and fixed throughout training ($\beta = 0$). We consider two settings: first, where the learning rate $\alpha$ is scaled as $\frac{1}{M}$ and $\sigma_M^2 = 1$ for all $M$, and second where $\sigma_M^2 = \frac{1}{M}$ and the learning rate $\alpha$ is equal to 1. These two settings yield the following dynamics, respectively:*

$$\lim_{M \to \infty} \partial_t \Phi_t^M \overset{P}{=} -(I - \gamma P^\pi)\Phi_t^M \quad , \text{ and} \tag{16}$$

$$\lim_{M \to \infty} \partial_t \Phi_t^M \overset{D}{=} -(I - \gamma P^\pi)\Phi_t^M + R^\pi \epsilon^\top \, , \epsilon \sim \mathcal{N}(0, I) \, . \tag{17}$$

*The corresponding limiting trajectories for a fixed initialisation $\Phi_0 \in \mathbb{R}^{\mathcal{X} \times d}$, are therefore given respectively by*

$$\lim_{M \to \infty} \Phi_t^M \overset{P}{=} \exp(-t(I - \gamma P^\pi))\Phi_0 \quad , \text{ and} \tag{18}$$

$$\lim_{M \to \infty} \Phi_t^M \overset{D}{=} \exp(-t(I - \gamma P^\pi))(\Phi_0 - (I - \gamma P^\pi)^{-1} R^\pi \varepsilon^\top)$$
$$+ (I - \gamma P^\pi)^{-1} R^\pi \varepsilon^\top \, , \epsilon \sim \mathcal{N}(0, I) \, . \tag{19}$$

One important corollary of this result occurs in sparse-reward environments under sub-optimal policies, where $R^\pi = \mathbf{0}$. In this case, we see that the representation converges precisely to the zero vector.

**Corollary 1.** *Let $\Phi_t^M$, $(w)_{i=1}^M$ be defined as in Theorem 1. Then if $R^\pi = 0$, the feature representation converges to the zero vector for every state, independent of whether the learning rate $\alpha$ is scaled as $\frac{1}{M}$ or the linear weight initialization variance scales as $\frac{1}{M}$. In particular:*

$$\lim_{t \to \infty} \lim_{M \to \infty} \Phi_t^M \overset{P}{=} \mathbf{0} \, . \tag{20}$$

*As a result, we have that the feature rank of $\Phi$ will also tend to zero*

$$\forall \epsilon > 0 \quad \lim_{t \to \infty} \lim_{M \to \infty} |\{\sigma \in SVD(\Phi_t^M) | \sigma > \epsilon\}| \overset{P}{=} 0 \, . \tag{21}$$

*Proof.* The proof of this result follows from a straightforward application of Theorem 1, setting $R^\pi = 0$ and letting $t \to \infty$. We can obtain an analogous result for the srank of $\Phi_t^M$ when $P^\pi$ is diagonalizable by noting that for any eigenvector $v_i$ of $P^\pi$, the value of $v_i^\top \Phi_t^M v_i$ evolves as $c \exp(-t\lambda_i)$ for some constant $c$ that depends on $\Phi_0^M$. In this case, we obtain a limiting value of 1 for the srank so long as $P^\pi$ corresponds to an ergodic Markov chain. $\qquad \square$

The setting of this result is distinct from that of deep neural network representation dynamics, as neural networks use discrete optimization steps, finite learning rates, and typically do not leverage linear ensembles. However, we emphasize two crucial observations that suggest the intuition developed in this setting may be relevant: first, in sparse reward environments the representation will be pushed to zero along dimensions spanned by the linear weights used to compute outputs. Once sufficiently many independent weight vectors are being used to make predictions, this effectively forces *every* dimension of the representation to fit the zero vector output. We would therefore expect representation collapse to be particularly pronounced in the QR-DQN agents trained on sparse-reward environments, as in this setting we obtain many independently initialized heads all identically trying to fit the zero target.

Second, in the presence of ReLU activations and stochastic optimization, the trajectories followed by the learned features in deep neural networks run the risk of getting 'trapped' in negative values. If these features would normally tend to small values close to zero (as we would expect in agents following similar dynamics to those obtained in Theorem 1), this increases the risk of unit saturation, where the representation may get trapped in bad local minima. This appears to be what happens in the QR-DQN agents trained on sparse-reward environments such as Montezuma's Revenge.

# B SEQUENTIAL SUPERVISED LEARNING

## B.1 DETAILS: TARGET-FITTING CAPACITY IN NON-STATIONARY MNIST

In addition to our evaluations in the Atari domain, we also consider a variant of the MNIST dataset in which the labels change over the course of training.

- **Inputs and Labels:** We use 1000 randomly sampled input digits from the MNIST dataset and assign either binary or random targets.

- **Distribution Shift:** We divide training into $N = 30$ or $N = 10$ iterations depending on the structure of the target function. In each iteration, a target function is randomly sampled, and the network's parameters obtained at the end of the previous iteration are used the initial values for a new optimization run. We use the Adam (Kingma & Ba, 2015) optimizer with learning rate `1e-3`, and train to minimize the mean squared error between the network outputs and the targets for either 3000 or 5000 steps depending on the nature of the target function.

- **Architecture:** we use a standard fully-connected architecture with ReLU activations, and vary with width and depth of the network. The parameters at the start of the procedure are initialized following the defaults in the Jax Haiku library.

We note that the dataset sizes, training budgets, and network sizes in the following experiments are all relatively small. This was chosen to enable short training times and decrease the computational budget necessary too replicate the experiments. The particular experiment parameters were selected to be the fastest and cheapest settings in which we could observe the capacity loss phenomenon, while still being nontrivial tasks. In general, we found that capacity loss is easiest to measure in a 'sweet spot' where the task for a given architecture is simple enough for a freshly-initialized network to attain low loss, but complex enough that the network cannot trivially solve the task. In the findings of the following section, we see how some of the larger architectures don't exhibit capacity loss on 'easier' target functions, but do on more challenging ones that exhibit less structure. This suggests that replicating these results in larger networks will be achievable, but will require re-tuning the task difficulty to the larger network's capacity.

## B.2 ADDITIONAL EVALUATIONS

We expand on the MNIST target-fitting task shown in the main paper by considering how network size and target function structure influences capacity loss.

- **Random-MNIST** (smooth) this task uses the images from the MNIST dataset as inputs. The goal is to perform regression on the outputs of a randomly initialized, fixed neural network. We use a small network for this task, consisting of two width-30 fully connected hidden layers with ReLU activations which feed into a final linear layer which outputs a scalar. Because the network outputs are small, we scale them by 10 so that it is not possible to get a low loss by simply predicting the network's bias term. This task, while randomly generated, has some structure: neural networks tend to map similar inputs to similar outputs, and so the inductive bias of the targets will match that of the function approximator we train on them.

- **Hash-MNIST** (non-smooth) uses the same neural network architecture as the previous task to generate targets, however rather than using the scaled network output as the target, we multiply the output by 1e3 and feed it into a sine function. The resulting targets no longer have the structure induced by the neural network. This task amounts to memorizing a set of labels for the input points.

- **Threshold-MNIST** (sparse) replaces the label of an image with a binary indicator variable indicating whether the label is smaller than some threshold. To construct a sequence of tasks, we set the threshold at iteration $i$ to be equal to $i$. This means that at the first iteration,

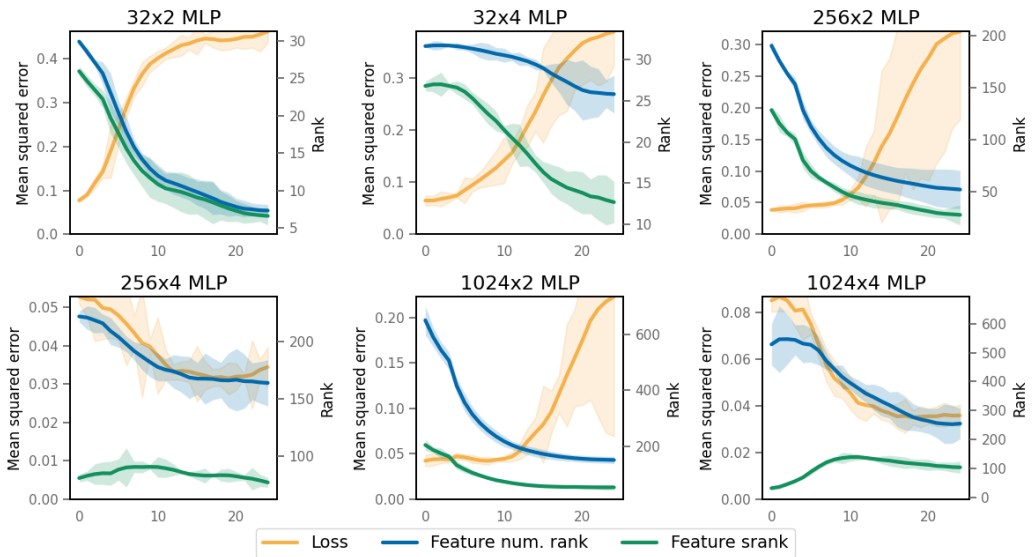

Figure 7: Mean squared error at the end of training on each iteration of the **hash-MNIST** task. Target-fitting error increases over time in smaller networks, but increasing the depth or width of the network slows down capacity loss, enabling positive transfer in the largest networks we studied.

the labels are of the form $(x, 0)$ for all inputs $x$. At the second iteration, they are of the form $(x, \delta(y < 1))$, where $y$ is the digit in the image $x$, and so on.

We consider MLP networks of varying widths and depths, noting that the network architecture used to generate the random targets is fixed and independent of the approximating architecture. We are interested in evaluating whether factors such as target function difficulty, network parameterization, and number of target functions previously fit influence the network's ability to fit future target functions. Our results are shown in Figure 7, 8, and 9. We visualize srank and feature rank of the features output at the network's penultimate layer, in addition to the loss obtained at the end of each iteration.

### B.3 EFFECT OF INFER ON TARGET-FITTING CAPACITY IN MNIST

In addition to our study of the Atari suite, we also study the effect of InFeR on the non-stationary MNIST reward prediction task with a fully-connected architecture; see Figure 10. We find that it significantly mitigates the decline in target-fitting capacity demonstrated in Figure 1.

## C ATARI EVALUATIONS

We now present full evaluations of many of the quantities described in the paper, along with a study of the sensitivity of InFeR to its hyperparameters. We use the same training procedure for all of the figures in this section, loading agent parameters from checkpoints to compute the quantities shown.

### C.1 HYPERPARAMETER SENSITIVITY OF INFER IN DEEP REINFORCEMENT LEARNING AGENTS

We report results of hyperparameter sweeps over the salient hyperparameters relating to InFeR, so as to assess the robustness of the method. For both the DDQN and Rainbow agents augmented with InFeR, we sweep over the number of auxiliary predictions (1, 5, 10, 20), the cumulant scale used in the predictions (10, 100, 200), and the scale of the auxiliary loss (0.01, 0.05, 0.1, 0.2). We consider

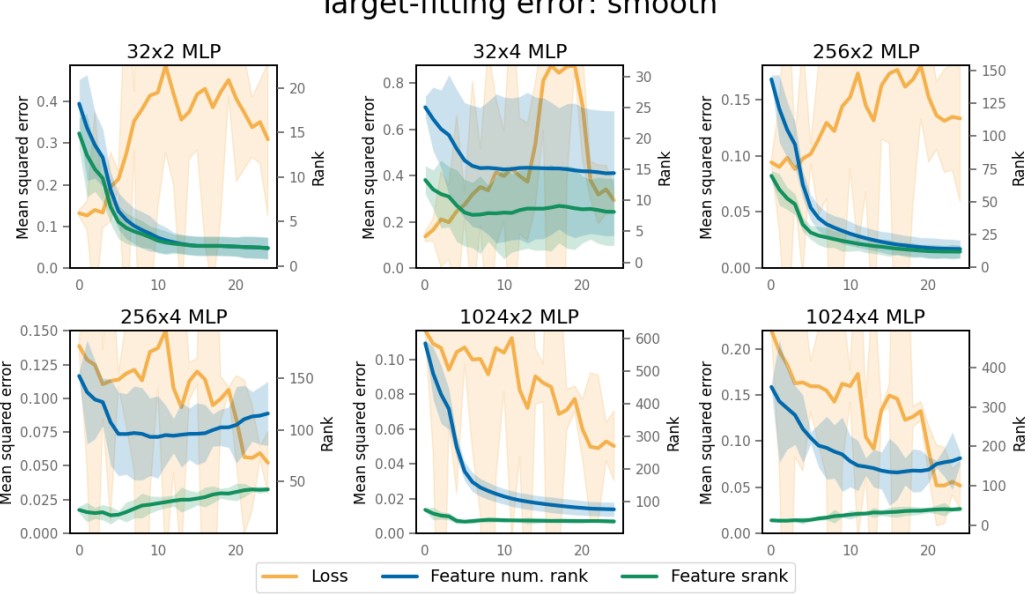

Figure 8: Mean squared error after 2e3 training steps on the **random-MNIST** task. Target-fitting error increases over time in under-parameterized networks, but increasing the depth or width of the network slows down capacity loss, enabling positive transfer in the largest network we studied.

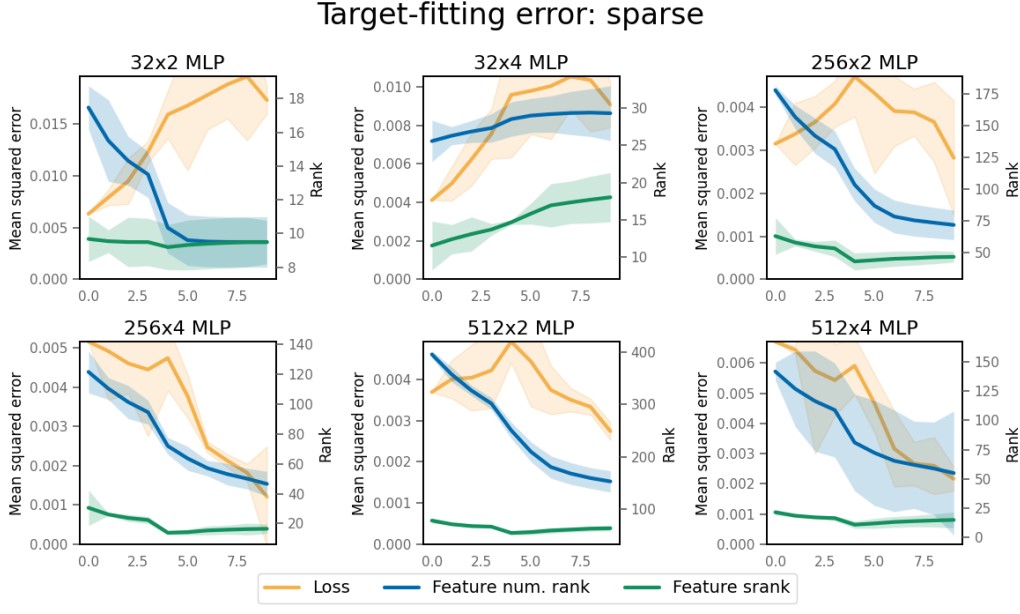

Figure 9: Mean squared error after 2e3 training steps on the **threshold-MNIST** task. Target-fitting error increases over time in under-parameterized networks, but increasing the depth or width of the network slows down capacity loss, enabling positive transfer in the largest network we studied.

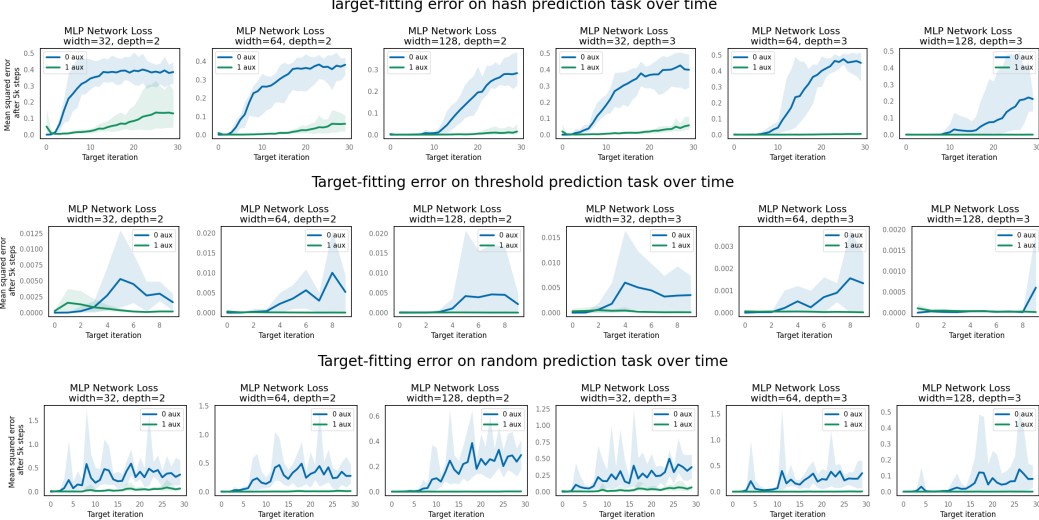

Figure 10: Effect of adding InFeR to the regression objective in a random reward prediction problem on the non-stationary MNIST environment studied previously. We see that the InFeR objective produces networks that can consistently outperform those trained with a standard regression objective, exhibiting minimal capacity loss in comparison to the same network architecture trained on the same sequence of targets.

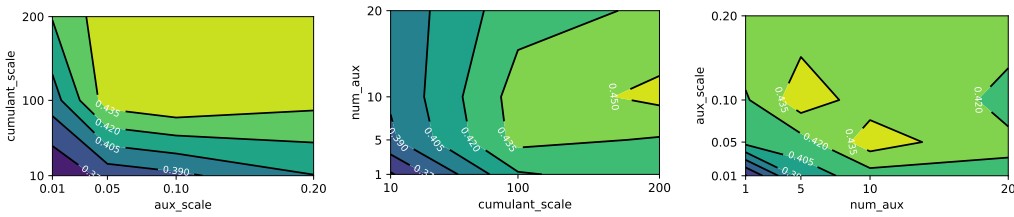

Figure 11: Hyperparameter sweeps for the DDQN+InFeR agent. Each contour plot shows average capped human-normalized score at the end of training marginalized over all hyperparameters not shown on its axes.

the capped human-normalized return across four games (Montezuma's Revenge, Hero, James Bond, and MsPacman), and run each hyperparameter configuration with 3 seeds. Results are shown in Figure 11 for the DDQN agent; we compare performance as each pair of hyperparameters varies (averaging across the other hyperparameter, games, and seeds, and the last five evaluation runs of each agent). Corresponding results for Rainbow are given in Figure 12.

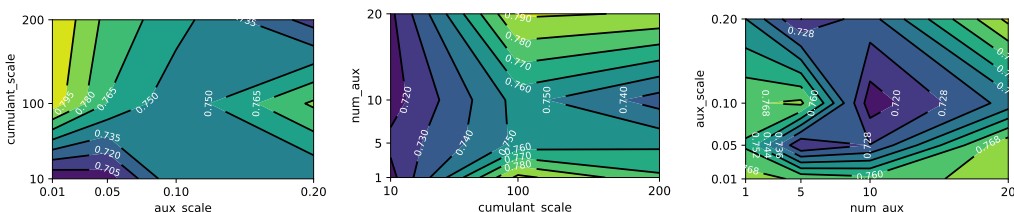

Figure 12: Hyperparameter sweeps for the Rainbow+InFeR agent. Each contour plot shows average capped human-normalized score at the end of training marginalized over all hyperparameters not shown on its axes.

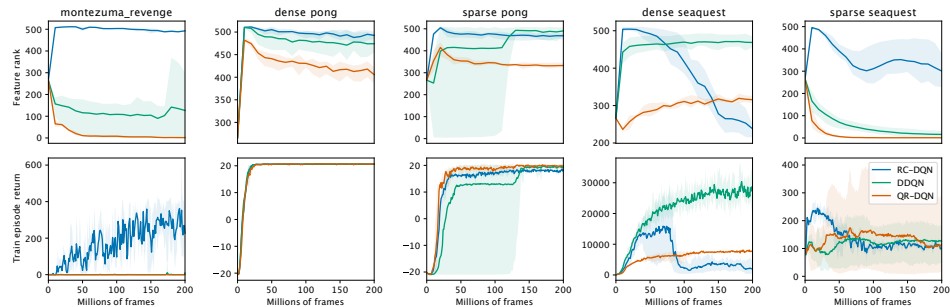

Figure 13: Feature rank and performance of RL agents on demonstrative Atari environments.

- **Agent:** We train a Rainbow agent (Hessel et al., 2018) with the same architecture and hyper-parameters as are described in the open-source implementation made available by Quan & Os-trovski (2020). We additionally add InFeR, as described in Section 4, with 10 heads, gradient weight 0.1 and scale 100.

- **Training:** We follow the training procedure found in the Rainbow implementation mentioned above. We train for 200 million frames, with 500K evaluation frames interspersed every 1M training frames. We save the agent parameters and replay buffer every 10M frames to estimate feature dimension and target-fitting capacity.

## C.2  FEATURE RANK

We first extend the results shown in Figure 3 to two additional games: Seaquest, and a sparsified version of Pong in which the agent does not receive negative rewards when the opponent scores. In these settings, we stored agent checkpoints once every 10M frames in each 200M frame trajectory, and used 5000 sampled inputs from the agent's replay buffer to estimate the feature rank, using the cutoff $\epsilon = 0.01$. Results are shown in Figure 13.

We further evaluate the evolution of feature rank in agents trained on all 57 games in the arcade learning environment. We find that the decline in dimension after the first checkpoint at 10M frames shown across the different agents in the selected games also occurs more generally in Rainbow agents across most environments in the Atari benchmark. We also show that in most cases adding InFeR mitigates this phenomenon. Our observations here do not show a uniform decrease in feature rank or a uniformly beneficial effect of InFeR. The waters become particularly muddied in settings where neither the Rainbow nor Rainbow+InFeR agent consistently make learning progress such as in tennis, solaris, and private eye. It is outside the scope of this work to identify precisely why the agents do not make learning progress in these settings, but it does not appear to be due to the type of representation collapse that can be effectively prevented by InFeR.

**Procedure.** We compute the feature rank by sampling $n = 50000$ transitions from the replay buffer and take the set of origin states as the input set. We then compute a $n \times d$ matrix whose row $i$ is given by the output of the penultimate layer of the neural network given input $S_i$. We then take the singular value decomposition of this matrix and count the number of singular values greater than 0.01 to get an estimate of the dimension of the network's representation layer.

In most games, we see a decline in feature rank after the first checkpoint at 10M frames. Strikingly, this decline in dimension holds even in the online RL setting where the agent's improving policy presumably leads it to observe a more diverse set of states over time, which under a fixed representation would tend to increase the numerical rank of the feature matrix. This indicates that even in the face of increasing state diversity, agents' representations face strong pressure towards degeneracy. It is worth noting, however, that the agents in dense-reward games do tend to see their feature rank increase significantly early in training; this is presumably due to the network initially learning to disentangle the visually similar states that yield different bootstrap targets.

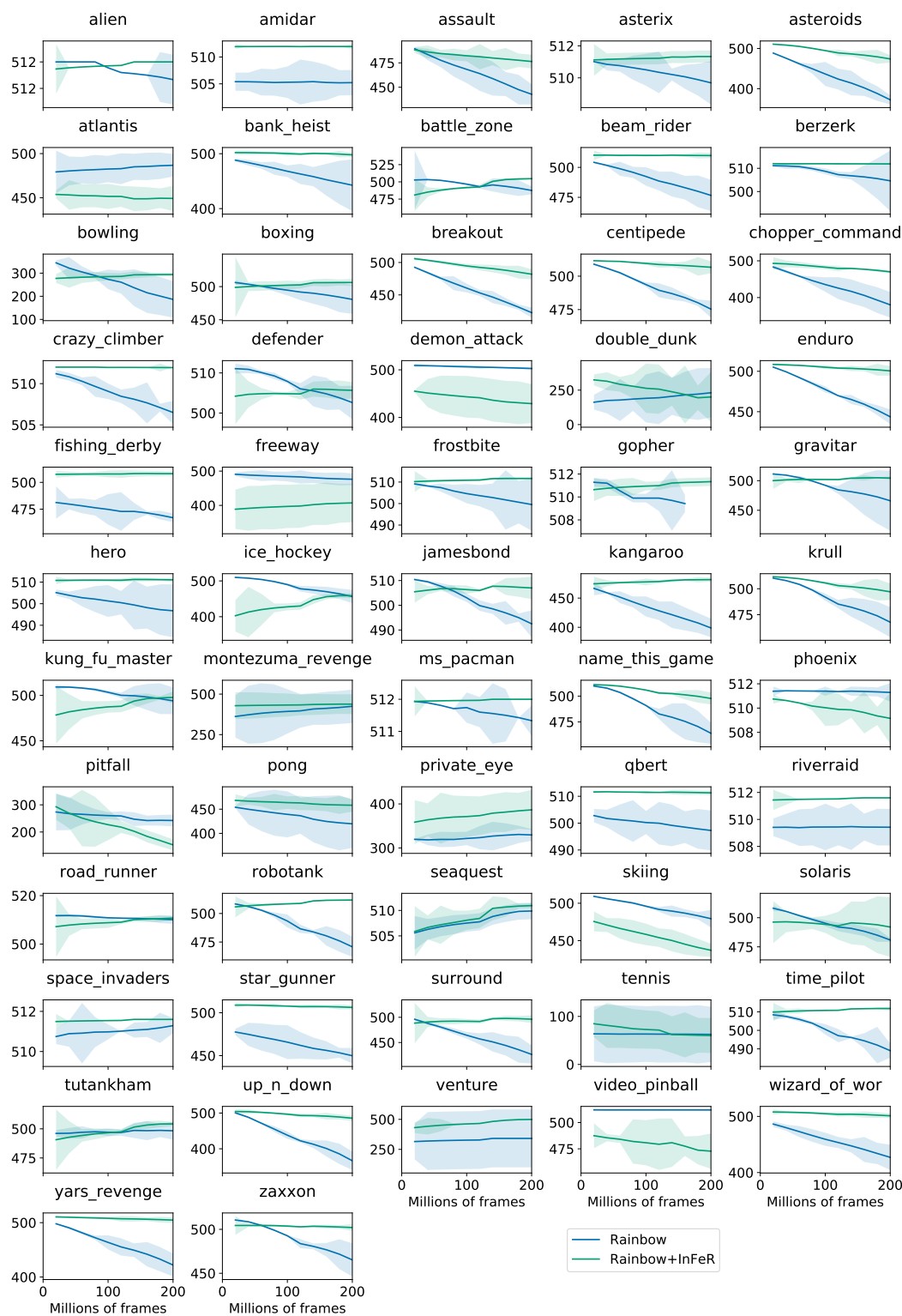

Figure 14: feature rank of agent representations over the course of training on all 57 games in the Atari benchmark. We compare Rainbow against Rainbow+InFeR. Rainbow+InFeR does not uniformly prevent decreases in feature rank across all games, but on average it has a beneficial effect on preserving representation dimension.

## C.3 TARGET-FITTING CAPACITY

In this section we examine the target-fitting capacity of neural networks trained with DQN, QR-DQN, and Rainbow over the course of 50 million environment frames on five games in the Atari benchmark (amidar, montezuma's revenge, pong, bowling, and hero). Every 1 million training frames we save a checkpoint of the neural network weights and replay buffer. For each checkpoint, we generate a random target network by initializing network weights with a new random seed. We then train the checkpoint network to predict the output of this random target network for 10000 mini-batch updates (batch size of 32) under a mean squared error loss, for states sampled from the first 100, 000 frames in the checkpoint's replay buffer. Furthermore, we repeat this for 10 seeds used to initialize the random target network weights.

The results of this experiment are shown in Figure 15 (in orange), where the solid lines show means and shaded regions indicate standard deviations over all seeds (both agent seeds (5) and target fitting seeds (10), for a total of 50 trials). We also show srank and feature rank of the features output at the network's penultimate layer for each of the checkpointed networks used for target fitting. These are computed using the network features generated from 1000 states sampled randomly from that checkpoint's replay buffer. For feature rank, averages and standard deviations are only over the 5 agent seeds.

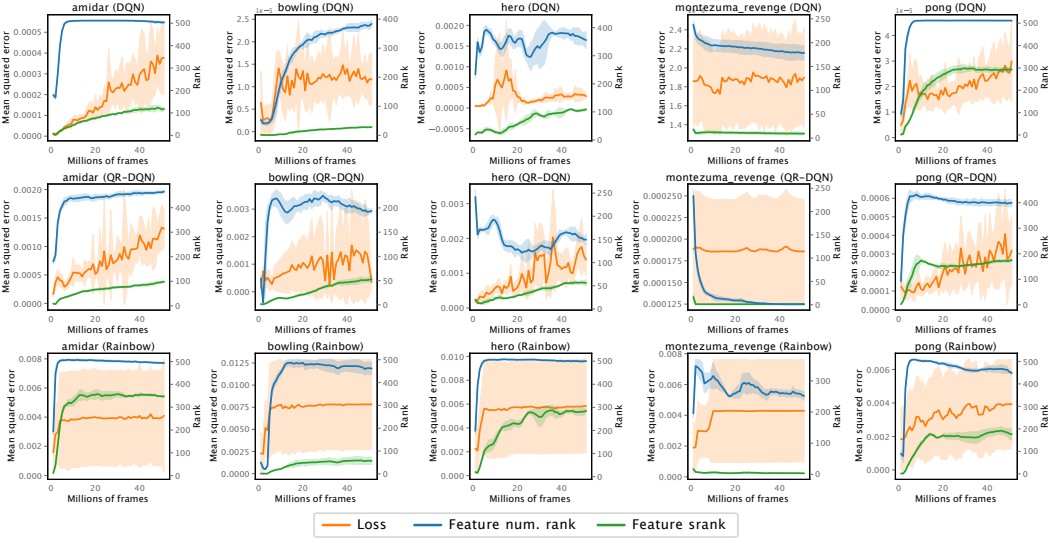

Figure 15: Mean squared error, after 10000 training steps for the target-fitting on random network targets. We also show the corresponding feature rank of the pre-trained neural network (before target-fitting).

## C.4 PERFORMANCE

We provide full training curves for both Rainbow and Rainbow+InFeR on all games in Figures 16 & 17 (capped human-normalized performance), and 18 & 19 (raw evaluation score). We also provide evaluation performance curves for DDQN and DDQN+InFeR agents in Figure 20.

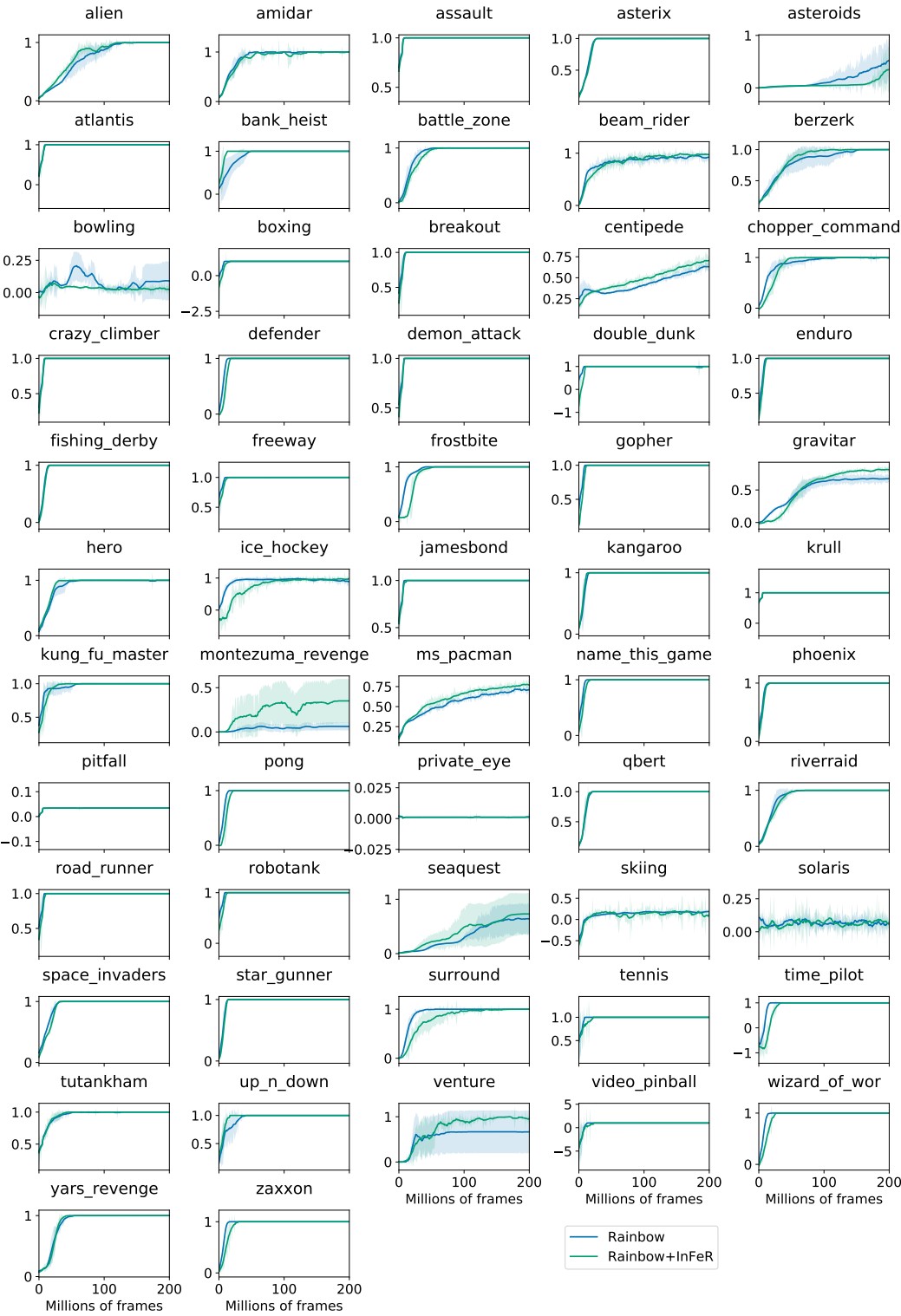

Figure 16: Full evaluation of capped human-normalized performance on Atari benchmarks for the default Rainbow architecture.

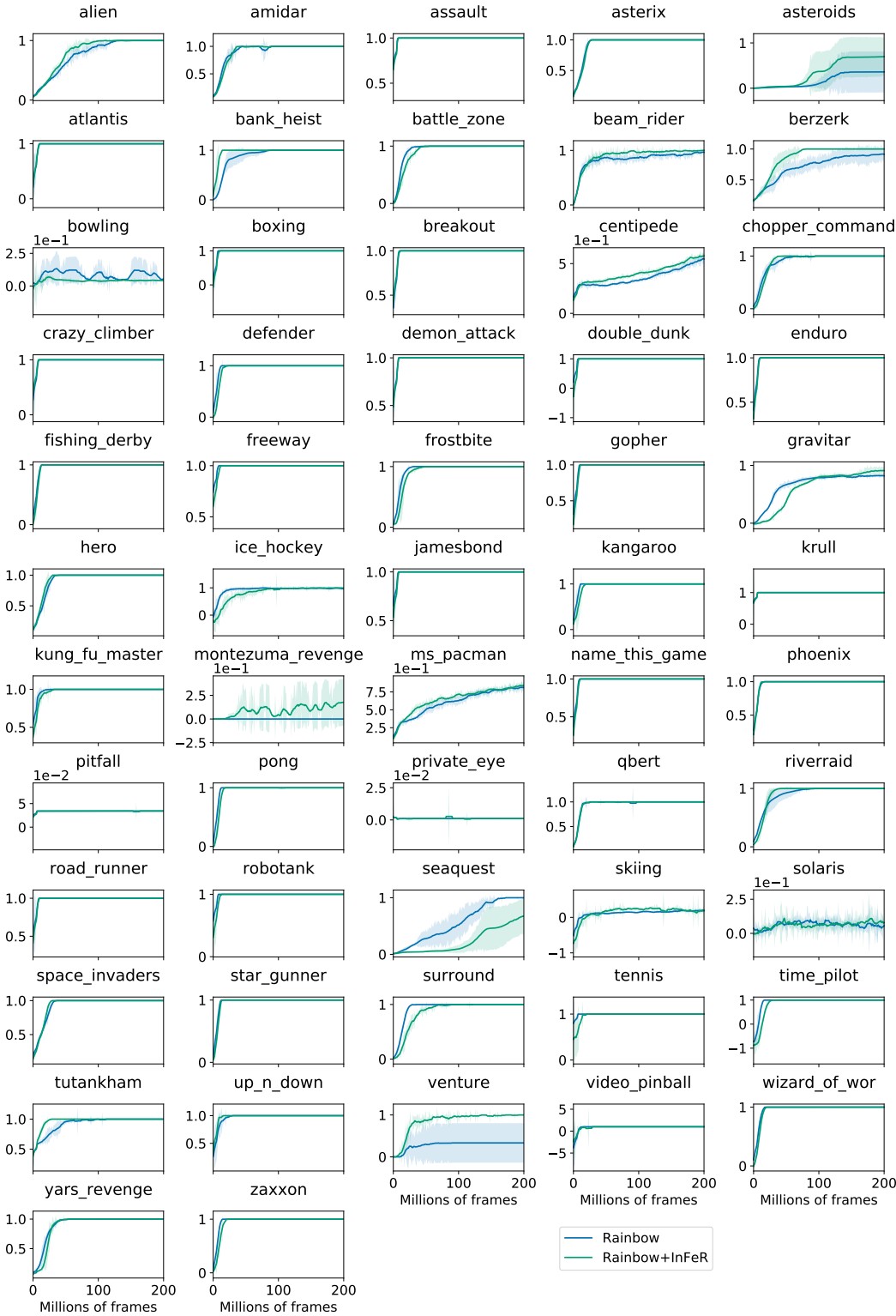

Figure 17: Full evaluation of capped human-normalized performance on Atari benchmarks in the double-width Rainbow architecture.

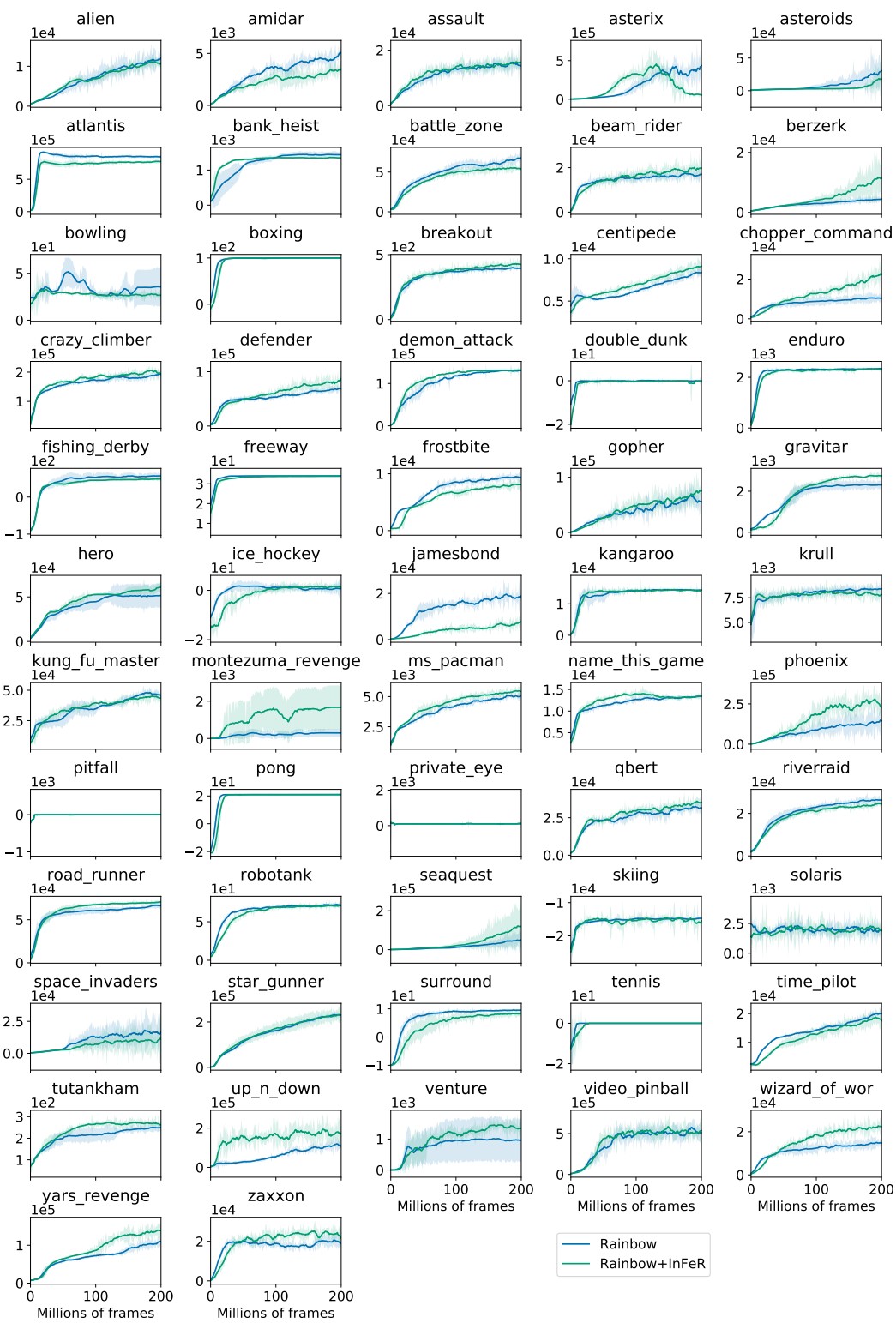

Figure 18: Full evaluation of raw scores on Atari benchmarks for the default Rainbow architecture.

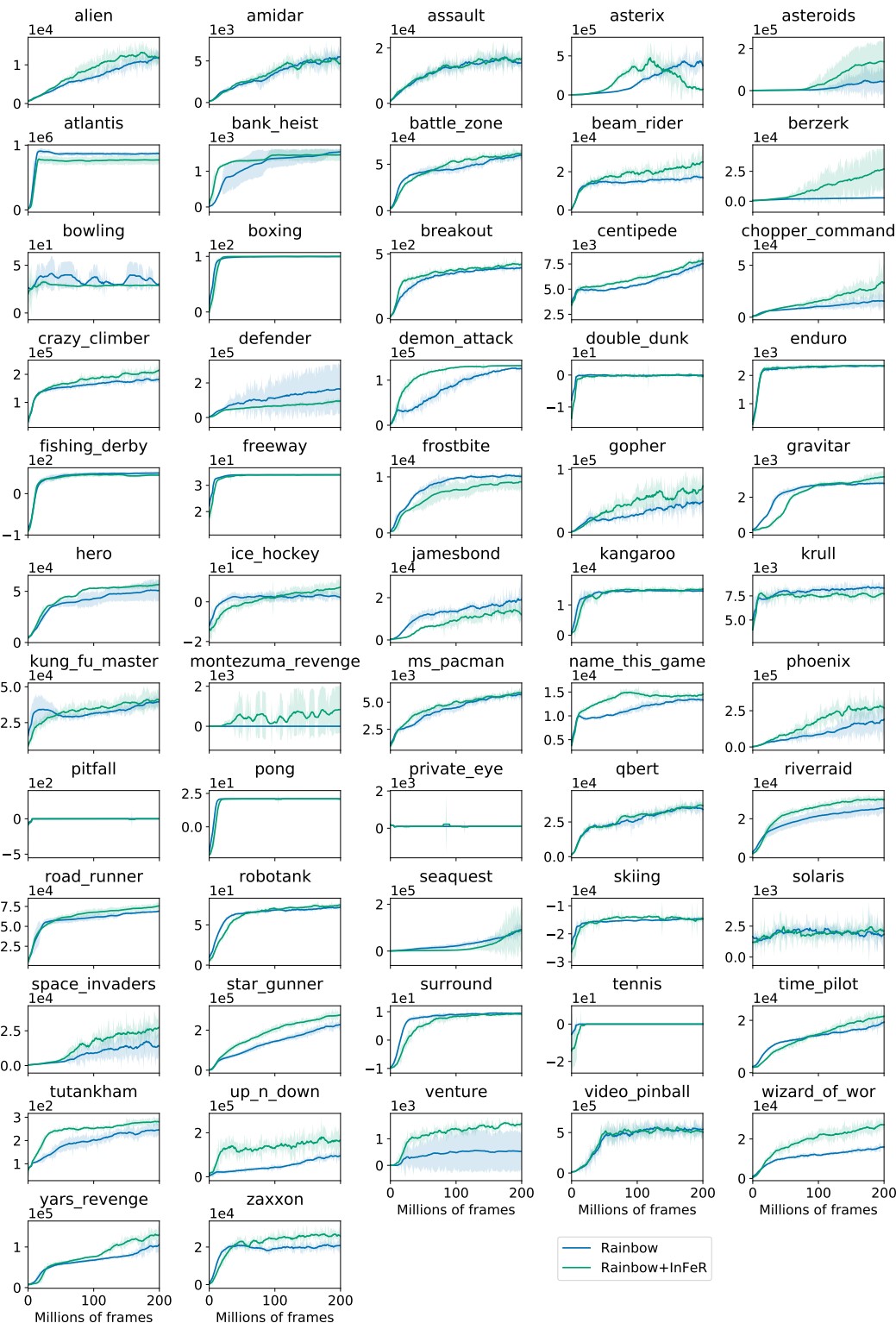

Figure 19: Full evaluation of raw scores on Atari benchmarks for the double-width Rainbow architecture.

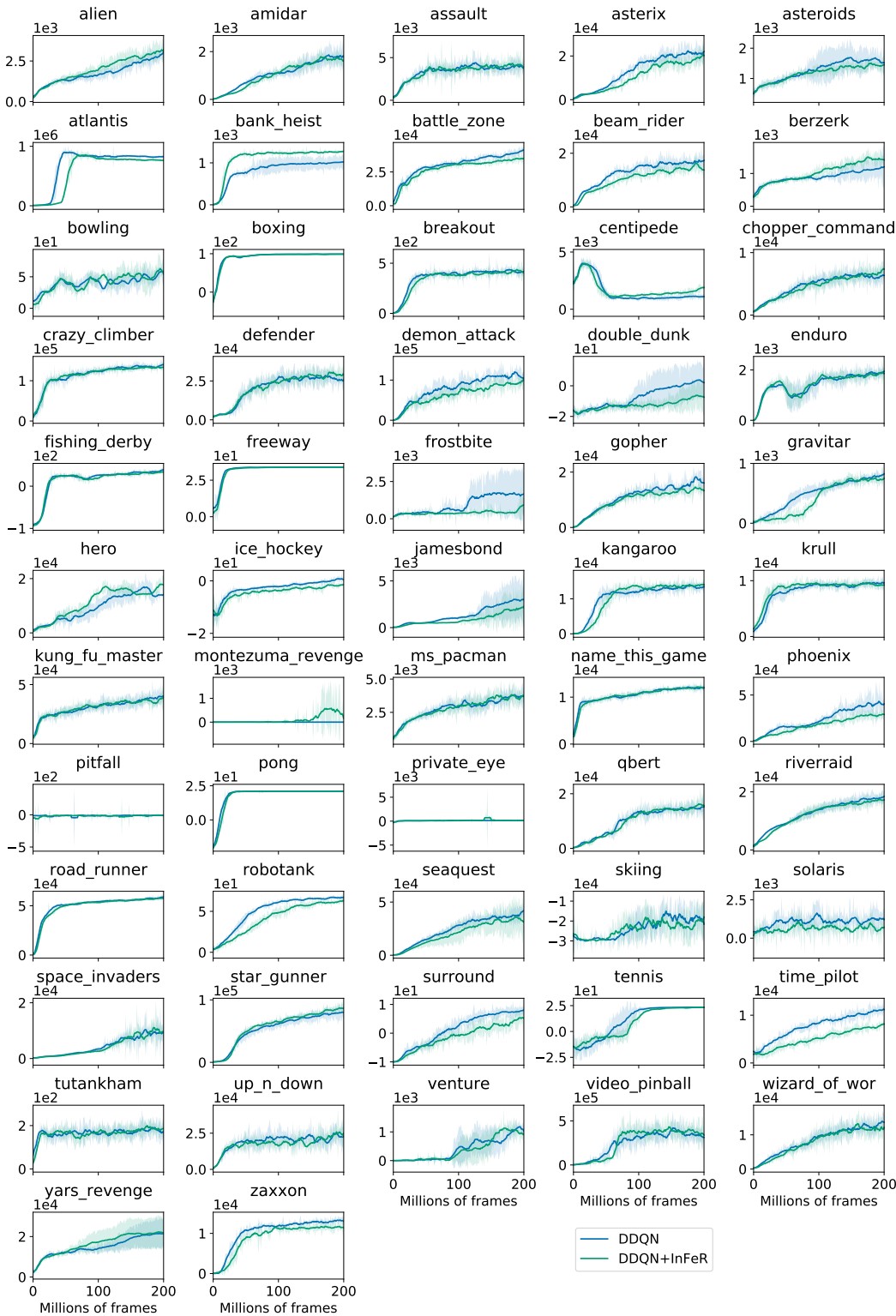

Figure 20: Evaluations of the effect of InFeR on performance of a Double DQN agent. Overall we do not see as pronounced an improvement as in Rainbow, but note that the average human-normalized score over the entire benchmark is nonetheless slightly higher for the InFeR agent, and that the performance improvement obtained by InFeR in Montezuma's Revenge is still significant in this agent.

