# OpenReview forum: "Understanding and Preventing Capacity Loss in Reinforcement Learning"
_ICLR.cc/2022/Conference — ICLR 2022 Spotlight_

### Official Review · Reviewer_vuGv · 2021-10-27

**Correctness:** 3
**Technical Novelty And Significance:** 2
**Empirical Novelty And Significance:** 2
**Recommendation:** 3
**Confidence:** 4

**Main Review:**

Strengths:

1. The paper tackles a super important question which is the stability of training in RL.

2. The high computational effort in running all Atari games for 200M frames.

Weaknesses:

3. The paper has no theoretical result, and even its capacity definition is thrown away quickly for a more practical measure.

4. Despite the heavy experimentations, I wasn't convinced by some of the plots, and some seem overbearing:
- Figure 1: Why use 3 algorithms to show the same point.
- Figure 2: Not sure I understood what is shown here besides the difference between activation functions - is this the point of the figure? because it seems a bit outside the scope of the paper.
- Figure 3: Does it make sense that at Frame count=0 not all plots start roughly at the same point - there wasn't training yet so why do we see a difference in the dimension? Also, we see the effective dimension can grow larger, if this is the case why would a collapse occur at all? Since the point in this Figure is to show a connection to sparsity, why not include a plot of the sparsity as well (like num steps where R!=0 divided by num steps)
- Figure 5 (b): The results for Robotank doesn't really follow the theme of the paper - both methods reach the same score even though one loses capacity. For Montezuma there is no capacity loss.
- Figure 5(c): The improvement does not look very consistent across games.

5. The proposed method of Infer looks a bit naive to me - why preserve random functionals and not for example an autoencoder of the state or some other meaningful feature. This method forces the network to keep random functions of the state instead of learning the underlying structure. You can also try to regularize for the rho-hat directly using some relaxation.


**Summary Of The Paper:**

The authors tackle a convergence issue in RL where the networks lose expressiveness during the training and are unable to regain it as the policy improves, ultimately leading to bounded performance. To solve this issue the authors propose adding a regularization term that forces the trained networks to regress to a random function of their initial output.

**Summary Of The Review:**

While the paper tackles an important problem, the proposed solution and it's empirical exhibition are insufficient in my view for acceptance.

---

> ### Author Response · Authors · 2021-11-17
> **Response to Reviewer vuGv (Part 1)**
>
>
> We thank the reviewer for their careful read of the paper, and for their comments, which highlight important directions to improve the presentation of our contribution. The points raised in this review are subtle, and highlight important points which we have endeavoured to clarify in our revisions to the paper. To summarize, we have shifted the narrative to clarify that the claim of the paper is not that capacity loss *uniquely* determines agent performance. Rather, it is that **under certain conditions** (such as when the network is underparameterized relative to the task at hand), **the sequential prediction problems faced in RL can lead to a measurable loss in capacity, and that in settings where network capacity is a bottleneck for performance, this leads to worse-performing agents**. We show that this occurs across many different environments in a popular benchmark, but do not claim that capacity loss is universally responsible for sub-optimal performance. For example, if an agent is significantly overparameterized relative to what is necessary to solve a task, then even large reductions in capacity may not have any effect on performance -- this is observed, for example, on the Robotank and Pong environments. We have implemented some updates to the text in Sections 1 and 3 to reflect this distinction more clearly.
>
> We now directly address specific concerns regarding the seeming inconsistencies between figures and the main message of the paper as follows.
> 1. Figure 5(b): "The results for Robotank doesn't really follow the theme of the paper - both methods reach the same score even though one loses capacity."
> We include results on Robotank to illustrate the diversity of environments in Atari: while some environments do seem to require agents to make use of the full capacity of the neural network, this is not the case across the board, particularly in “easy” environments where capacity is not a bottleneck to performance. For example, both Pong agents and Robotank agents exhibit modest declines in capacity over the course of training, but the agents' performance remains relatively unaffected by this capacity loss. Importantly, we do not claim that capacity loss is the only, or even in many cases the most important, factor influencing performance. Other properties of the agent’s representation, exploration strategy, and loss function can also play a role.
> 2. “For Montezuma there is no capacity loss.”
> Montezuma’s Revenge presents an interesting case study. The Rainbow agent doesn’t exhibit a notable decline in its ability to fit random network targets between 10M and 200M frames, but does exhibit markedly lower effective dimension compared to when InFeR is also used. This can be attributed different behaviour early in training; our results in Appendix B demonstrate that a distributional agent in particular exhibits **rapid representation collapse well before the first checkpoint**, with notable divergence from a random initialization even within the first 2 million training steps. Given that our checkpointing started after 10 million frames, the apparent lack of capacity loss in Montezuma’s Revenge can be largely attributed to the coarseness of our analysis. We will develop and cite these preliminary results (the placeholder figure in the Appendix is merely meant to be illustrative and will be updated in the coming days) in the main body of the paper to clarify this issue.
>  3. "Figure 5c):The improvement does not look very consistent across games.“
> As discussed previously, we would not a priori expect InFeR to uniformly affect performance in all games relative to the baseline unless we also believed that capacity loss uniformly limits performance in all games. In a simple task such as Pong, for example, agents quickly attain an approximately optimal policy, after which point the agent only needs to predict a single, relatively simple, value function which does not require the full network capacity to represent.
>
> 4. “why preserve random functionals and not for example an autoencoder of the state or some other meaningful feature.”
> We focus on random functionals so as to isolate the effect of capacity loss on learning progress. Using a more meaningful representation would introduce additional complexities into our evaluation and make it difficult to disentangle the effect of the auto-encoder objective on more complex properties of the representation as opposed to solely its effect on capacity.

---

> ### Author Response · Authors · 2021-11-17
> **Response to Reviewer vuGv (Part 2)**
>
>
> 5. “ You can also try to regularize for the rho-hat directly using some relaxation.”
> This is an interesting approach which would resemble that employed by Agarwal et al. in the offline setting. Such an approach would certainly have a beneficial effect on the effective dimension of the representation. Because we try to remain agnostic to the particular notion of capacity used, and as we have discussed previously, effective dimension and capacity loss measure subtly different quantities, we focus on an objective that is relevant to many notions of capacity. In particular, it is not obvious how preserving the srank of the representation would affect the ability of the network to fit arbitrary targets. In contrast, the InFeR regularization  explicitly preserves the capacity of the network to output functions it was able to represent at initialization. The effect of dimensionality-focused regularization schemes on other notions of capacity will be an interesting direction for further study, but falls outside the scope of the current paper.
>
> 6. “The paper has no theoretical result”
> The motivation for this paper is grounded in the theoretical results of Lyle et al. (2021). In particular, we note in sparse-reward settings that these results suggest that the representation may collapse along particular dimensions which depend on the linear function approximator used in the network architecture. We have included a more detailed discussion of the implications of these results for our setting, including a novel corollary concerning feature collapse in sparse-reward environments in Appendix A.
> 7. "Figure 1: Why use 3 algorithms to show the same point.""
> The point being shown in Figure 1 is that capacity loss occurs under many different learning rules; it is not a phenomenon specific to a single RL algorithm, but seems to occur robustly across both expected and distributional agents, and even in the presence of auxiliary tasks. The use of 3 algorithms is to emphasize that this is a widely-occuring phenomenon.
>
> 8. "Figure 2: Not sure I understood what is shown here besides the difference between activation functions - is this the point of the figure?"
> The figure highlights two critical points: first, that neural networks which have been trained to fit a sequence of different target functions see reduced ability to quickly fit new target functions. This is demonstrated by the fact that the lines corresponding to the MSE of a range of network architectures increase as the number of tasks seen by the network increases (with a caveat that in the binary prediction setting the later tasks are easier than the middle task). The second point made by figure 2 is that the mean squared error on the prediction task increases by a much larger absolute value for the smaller network, suggesting that this effect will be particularly exacerbated in network architectures that are under-parameterized relative to their prediction task. These insights are useful for interpreting the Atari 2600 results, where the network architecture is fixed, but the difficulty of prediction problem corresponding to the different games can vary dramatically.
>
> 9. "Figure 3: at Frame count=0 not all plots start roughly at the same point"
> Because we load checkpoints starting from 10M training frames (the checkpoints at t=0 do not have any data in their replay buffers on which to evaluate capacity), the plots do not in fact start from t=0, but rather t=10M. The first 10M frames of training are omitted due to limited checkpointing capacity. We have begun evaluations to provide more insight into this early training period, a preliminary version of which can be seen in Appendix B.

---

> ### Author Response · Authors · 2021-11-17
> **Response to Reviewer vuGv (Part 3)**
>
>
> 10. "Also, we see the effective dimension can grow larger, if this is the case why would a collapse occur at all?"
> We measure two distinct notions of capacity: target-fitting capacity and effective dimension. As we are not sure which the reviewer is referring to with this point, we include responses to both.
> The **effective dimension** depends on a complex set of competing and confounding factors: increasing the diversity of the state visitation distribution, large variance in the value function, and other properties might lead the effective dimension to increase over training. Sparse rewards and low variability in the state visitation distribution tend to decrease the effective dimension. In dense-reward environments, the agent receives sufficient learning signal so that it can distinguish states of different values, whereas in sparse-reward environments where all inputs point to similar targets, effective dimension tends to be quite low, particularly in QR-DQN agents, where the agent must fit many quantile predictions to this zero target. Thus, the answer to the reviewer’s question is that effective dimension can increase when the agent receives a rich learning signal from the environment, but when the network is not incentivized to disentangle states, such as in sparse-reward environments, the representation will often collapse. Our empirical results support this conclusion: in particular, Figure 4 a) highlights low effective dimension in agents which attain zero returns, with “full rank” representations predominantly occurring in agents which obtain non-trivial rewards.
>     **Target-fitting capacity** will similarly benefit from an agent’s ability to distinguish states in the environment. Thus, early in training and in agents with sufficient capacity, it might be that the early training dynamics encourage agents to distinguish states which have different rewards, resulting in positive transfer to a random target-fitting task. Similarly to the effective dimension, however, if the agent begins to map different states to the same feature or saturates some ReLU units, this notion of capacity may also begin to decline. This is roughly what we observe in the nonstationary MNIST experiments: in under-parameterized models, capacity decreases over time, while in over-parameterized models there can be some positive transfer between tasks early in training.

---

> ### Comment · Reviewer_vuGv · 2021-11-22
> **Following authors comments**
>
> I thank the authors for their detailed responses and comments.
>
> I still think the paper deals with a very important issue, but the evidence still feels circumstantial and the case in my view is not strong enough for publication:
>
> 1. Some environments \ agents lose capacity, some don't.
>
> 2. When capacity is lost, it can be regained. Which doesn't really follow the intuition about the term "capacity loss".
>
> 3. Sometimes low capacity affects performance, sometimes it doesn't.
>
> 4. Sometimes high capacity does not lead to good performance, sometimes it does.
>
> 5. The proposed way to prevent capacity loss is to keep the ability to represent predetermined randomized functions. These random functions are unrelated to the environment, dynamics, capacity definition, or reward, which I think reflects on how the capacity loss is actually not well understood.
>
> I'm keeping my score as I think the work is not mature enough for publication.

---

> > ### Author Response · Authors · 2021-11-25
> > **Clarifying the Paper's Empirical Claims (Pt 1)**
> >
> > We thank the reviewer for engaging in the discussion period, and for elaborating on their barriers to recommending acceptance. Based on this response, we suspect the reviewer’s concerns stem from a lack of clarity on our part in explicitly laying out the specific empirical claims the paper is making, rather than discordance between our claims and our empirical results. We recall from Section 1: “The principal thesis of this paper is that over the course of training, deep RL agents lose some of their capacity to quickly fit new prediction tasks, and in extreme cases this capacity loss prevents the agent entirely from making learning progress”. This thesis has two main nuances:
> >
> > 1. We claim that low capacity blocks learning progress. We do not claim that high capacity is sufficient to guarantee good performance. We agree with the reviewer that our empirical results would not support such a claim, which is why we do not make it in the paper; we will include a discussion of this in Section 3.2 to make this clearer.
> >
> > 2. In Section 3, we lay out specific conditions which result in capacity loss in deep neural networks: non-stationary targets and under-parameterization relative to the target task. These conditions are frequently satisfied in deep RL settings, resulting in capacity loss. We see now how the quoted sentence can be interpreted as stating that capacity loss is a significant problem in all deep RL agents/environments, which was not its intended meaning. We will update this in future revisions as follows: “...over the course of training, non-stationary TD targets and sparse rewards can cause deep RL agents to lose some of their capacity to quickly fit new prediction tasks…”
> >
> > We also emphasize a key empirical finding of our paper: that by using InFeR in combination with epsilon-greedy exploration, it is possible to obtain performance on Montezuma's Revenge competitive with agents using more advanced exploration techniques such as noisy nets [1] and count-based exploration [2], while still maintaining good performance in dense-reward games. The following paragraphs in the next response provide more detailed responses to specific concerns.
> >
> > [1] Fortunato, Meire, et al. "Noisy Networks For Exploration." International Conference on Learning Representations. 2018.
> >
> > [2] Bellemare, Marc, et al. "Unifying count-based exploration and intrinsic motivation." Advances in neural information processing systems 29 (2016): 1471-1479.

---

> > ### Author Response · Authors · 2021-11-25
> > **Addressing specific comments (Pt 2)**
> >
> >
> > > Some environments \ agents lose capacity, some don't.
> >
> > This is addressed by bullet point 2 above, which clarifies the settings under which we expect to be able to measure capacity loss. Further, pointing to Figures 10 and 11 we see that the vast majority of Rainbow agents exhibit declining effective dimension and increasing target-fitting error over the course of training. This suggests that capacity loss is a relevant phenomenon to consider to understand deep RL. The environments from Figures 10 and 11 where we don’t see capacity loss tend to either see high capacity throughout all of training, or take on low values within the first 10M frames, suggesting that capacity loss occurred too early in training for our checkpointing to detect. This claim is confirmed  in the agent trained on Montezuma’s Revenge in Figure 7.
> >
> > > When capacity is lost, it can be regained. Which doesn't really follow the intuition about the term "capacity loss".
> >
> > This concern seems to stem from different interpretations of “capacity”, and whether it makes sense for a definition of capacity loss to be reversible. Our definition of network capacity is specifically geared towards the network’s ability to perform the kinds of updates that lead to short-term performance improvements. This means a network which has lost capacity hasn’t necessarily lost its ability to ever learn again, but rather that its current parameter values are in an unfriendly region of the optimization landscape from which it is difficult to perform updates that lead to policy improvement, making this definition well-suited to RL.
> >
> > We assume the reviewer is specifically noting the example of DQN in Sparse Pong in this comment. While this might seem like an exception to the capacity loss narrative, in that the network’s representation initially collapses and then recovers, it is consistent with our observations that a) uniformly zero rewards decrease network capacity, and b) if a network is exposed to suitably informative learning signals, its representation will adapt to distinguish states. Importantly, this results in the following causal chain: a shift in the observed reward signal -> increased network capacity -> improved performance. Importantly, the increase in capacity precedes the improvement in performance: we care specifically about the agent’s ability to improve its performance and this shows that our notion of capacity is accurately measuring this property.
> >
> > > Sometimes low capacity affects performance, sometimes it doesn't.
> >
> > We have run extensive empirical evaluations and did not see any examples of agents which made learning progress while their representation had collapsed. This is consistent with our claim that extreme capacity loss prevents learning progress. As explained earlier, we don’t make any claims about a direct linear relationship between capacity and performance outside of this regime.
> >
> > > Sometimes high capacity does not lead to good performance, sometimes it does.
> >
> > Again, we do not make any claims about high capacity being sufficient for good performance.
> >
> > > The proposed way to prevent capacity loss is to keep the ability to represent predetermined randomized functions. These random functions are unrelated to the environment, dynamics, capacity definition, or reward, which I think reflects on how the capacity loss is actually not well understood.
> >
> > While we do not explicitly draw this connection in the paper, the proposed regularizer is indeed directly motivated by the notion of capacity we propose. Our measure of capacity evaluates the ability of a network to fit random functions: by sampling random functions and preserving the network’s ability to fit them, we are regularizing the network towards parameters which can represent an empirical distribution of functions from this class. With respect to feature rank, the InFeR objective acts so as to enforce a soft lower bound on the effective dimension: because the network must preserve the linear outputs, it cannot collapse its representation too far along the random subspace spanned by the corresponding weights. In principle, any set of target functions could be used for regularization, depending on the function class used to define capacity.
> >
> > We point the reviewer to Appendix A, where we discuss how the learning dynamics induced by TD updates already encourage agents to align their representation to the principal components of the environment transition matrix; preserving random functionals on top of this is designed to prevent representation collapse while still following the TD dynamics along the orthogonal complement to these weights. This justification was not included in our original draft due to limited space, but we have fleshed out the learning dynamics connection more in Appendix A of our revised version.
> > Finally, we point to the updated Figure 5 where we show that InFeR mitigates capacity loss in the non-stationary task used to illustrate capacity loss in Figure 1.

---

> > ### Author Response · Authors · 2021-11-30
> > **Please see discussion with reviewer Mqcv if additional concerns remain**
> >
> > Thanks to the reviewer for their engagement in the discussion period. As the discussion period is ending today, we will not be able to respond to any follow-up comments the reviewer may have. We refer the reviewer to our discussion with reviewer Mqcv if our response has flagged any additional questions, as this discussion revealed a number of subtle but important aspects of our contribution and analysis that may not have been made clear in our initial response.

---

### Official Review · Reviewer_Mqcv · 2021-10-30

**Correctness:** 3
**Technical Novelty And Significance:** 3
**Empirical Novelty And Significance:** 3
**Recommendation:** 6
**Confidence:** 4

**Main Review:**

==========================

Strengths:
- novelty: this problem is not adequately studied in the RL setting, so I believe the authors are making some interesting contribution
- interesting empirical analysis and insights are provided.
- empirical result: the result on Montezuma's revenge show that the proposed method (which is very simple) can improve performance a lot, which seems to support the claims of the paper.
- writing quality: good

==========================

Weaknesses:

Major concern:

Figure 4 (b) I'm a bit concerned whether this figure actually supports the argument that agent gradually lose capacity during training. Authors claim that this figure shows that learning happens only after agent *recovers* from representation collapse. Note that Figure 1, 2 show that agent gradually lose capacity during training. But figure 4 (b) shows this agent's capacity is "collapesed" right from the very beginning of trainining, and then later this capacity is *significantly increased*. To make it more clear:

1) Figure 1, 2 seem to show capacity (measured with MSE) starts *high* then gradually get *lower*

2) Figure 4 (b) seems to show capacity (measured with effective rank) starts "low" then can get "high"

So it seems to me these 2 conclusions are actually... disagreeing with each other? Would be great if you can address this concern.

An potentially interesting experiment is... if InFeR is added to middle of training that sparse pong unlucky seed, should we see the capacity immediately increase? To be specific, I'm talking about comparing 2 curves, the first curve trains that unlucky seed as usual, the second curve uses the same seed, but when trains to sth like 50M frames, InFeR is added. In this case, will we see InFeR immediately increase the effective rank, and potentially immediately after bring performance up?


Other concerns and comments:

- One thing seems a bit lacking to me is comparison to other methods that improve representation learning. Have you try compare InFeR with any other methods that center around representation learning? For example, contrastive learning methods like CURL, will they achieve the same capacity-preserving effect? What is the advantage of using InFeR compared to other representation learning methods?

- A disucssion on the potential negative effect of the proposed regularization scheme would be good: in some environments the performance dropped a bit after adding INFER, what could be the reason? And if the initial values are simply not good enough, will this method lead to performance drop?

- Page 4 when authors explain Figure 2, "We further consider a ‘sparse-reward’ version of MNIST..." after this there is one sentence to explalin the MNIST experiment setting, and this seems a bit too concise. I can still get what you mean, but would really prefer you try to expand the explanation a bit and maybe give an example or schematic, you can put this in the appendix if needed, it would be really helpful for readers to understand it fast and prevent potential confusion.

- page 1 in introduction, some very recent results on data augmentation (mainly DrQ and DrQv2, and this is also used in CURL, temporal augmented contrast etc.) actually find that data augmentation can boost RL performance quite robustly. Perhaps you want to consider rewrite this part slightly?

**Summary Of The Paper:**

Summary of paper:
- The authors identify a problem of "capacity loss" during RL training with neural networks, this is the problem of agents gradually losing the ability to fit new functions while trainig.
- The authors argue this is a key factor that hinders learning and is most prominent in sparse reward environments, propose 2 different measures of capacity loss and conduct empirical analysis to show this phenomenon exist in some Atari tasks.
- The authors present a number of empirical analysis on this problem, and propose a simple regularization scheme to mitigate this issue, and presented some analysis on the effect of this scheme, giving interesting insights.
- The proposed method improve performance especially on Montezuma's revenge when no specialized exploration techniques are used, supporting the paper's claims

**Summary Of The Review:**

Overall I think the paper makes some interesting and novel contributions, the problem of capacity loss is rarely studied in the RL setting, and research in this direction might help us better understand the difficulty in DRL training. Though I currently do have some concerns. I look forward to reading the authors' rebuttal. I will consider increase my score if my concerns are fully addressed.

========================================================

post-rebuttal

I now increase my score to 6 since I feel a lot of the concerns are indeed addressed in the response. However the authors should go through the paper carefully and try to make things more clear in the paper, would be nice to incorporate some of these reviewer discussions into the paper or appendix.

---

> ### Author Response · Authors · 2021-11-17
> **Response to Reviewer Mqcv**
>
> We thank the reviewer for their careful read of our paper and look forward to engaging in the discussion period. We have endeavoured to address many of the more straightforward points in our initial update to the pdf of the paper and supplemental material to improve the clarity of the paper and to resolve non-obvious ambiguities in our empirical results. The review has highlighted a number of nuanced points which, while not contradictory to our principal claims, do shed light onto some non-obvious properties of the measures we use to evaluate capacity in neural networks and have highlighted areas of the paper where our explanation was not sufficiently clear. We include responses to specific comments below.
>
> 1. “Figure 4 (b) I'm a bit concerned whether this figure actually supports the argument that agent gradually lose capacity during training.”
> The reviewer is right to point out that the figures measuring the target-fitting capacity of a network trained on Montezuma’s Revenge in Figure 4(b) are relatively flat. However, the value of the effective dimension taken by each random seed looks quite different from that of the randomly initialized network, and so the apparent disagreement with our claims can be largely attributed to the coarse checkpointing we performed in our initial experiments. We include evaluations of distributional agents in Appendix B which show that much of the decline in effective dimension which occurs in Montezuma's Revenge happens in the first 10M frames, before the first checkpoint has been logged, against a randomly initialized baseline. We see that capacity loss in montezuma's revenge occurs early in training, and plan to include a more thorough investigation into these agents. We will post an update when we have added these more thorough results to the supplementary material.
>
> 2. Comparisons with other representation learning methods: we agree that studying other representation learning methods through the lens of capacity loss is an important and exciting direction for further work. Given the difficulty of isolating the effect of capacity loss as opposed to other properties of a representation, we have refrained from studying the effect of representation-learning methods on capacity loss; developing the empirical and theoretical tools necessary to disentangle the effect of these objectives on non-capacity properties of the representation from their effect on our generic formulation of capacity is outside of the scope of this work, which focuses on standard RL objectives. Our motivation for proposing InFeR is to isolate the effect of representation collapse independent of other properties of the representation, such as its ability to reconstruct rewards or predict future observations, by considering an objective which only directly influences the dimensionality of the representation without incorporating additional environment information. Because InFeR is simply performing feature space regularization, it is complementary to any additional representation-learning loss that one could add to an agent. We are excited to study how the ideas presented in this paper interact with representation-learning methods as future work, but to do this analysis properly would be outside the scope of the current paper.

---

> > ### Comment · Reviewer_Mqcv · 2021-11-28
> > **Response to rebuttal**
> >
> > First thank you for the rebuttal and the clarification, and sorry for the late reply.
> >
> > I feel that the authors' reponse addressed a lot of my concerns. Thank you for the effort. However, now the paper seems to be giving a list of "circumstantial" (from Reviewer vuGv) observations that are not that consistent over different tasks and other factors. (This not necessarily means the paper is not interesting, but) It seems to me the main conclusions of the paper is now becoming a little hard to grasp.
> >
> > And one thing that still bothers me is this (quoting from your reply to reviewer vuGv)
> > > We assume the reviewer is specifically noting the example of DQN in Sparse Pong in this comment. While this might seem like an exception to the capacity loss narrative, in that the network’s representation initially collapses and then recovers, it is consistent with our observations that a) uniformly zero rewards decrease network capacity, and b) if a network is exposed to suitably informative learning signals, its representation will adapt to distinguish states. Importantly, this results in the following causal chain: a shift in the observed reward signal -> increased network capacity -> improved performance. Importantly, the increase in capacity precedes the improvement in performance: we care specifically about the agent’s ability to improve its performance and this shows that our notion of capacity is accurately measuring this property.
> >
> > I tried to think through this but feel that I am still a little confused. Say that we are in a sparse reward environment, and then after the agent gets some new data that happen to have reward, and then the agent now has **reward signal**, and then it has **increased capacity**, and then **performance improvement**. That is indeed what the paper observed, however, if the agent can simply increase capacity when it gets new data with reward, then is this kind of capacity loss really a problem? How exactly is this different from an exploration problem? In the abstract, I believe the following sentence was meant to show the main argument of this paper
> >
> > > We demonstrate that capacity loss occurs in a broad range of RL agents and environments, and is particularly damaging to learning progress in sparse-reward tasks
> >
> > Now, if agents can just get new data and then recover capacity, then, is capacity loss really **damaging performance**?
> >
> > Let me try to be really clear what I meant:
> >
> > 1. I feel that the paper is trying to make a claim that certain ways of training (for example, no INFER and in sparse reward env) will lead to the issue of **capacity loss**, and this problem further leads to **bad performance**. Thus to me seems the main story of the paper is **typical way of training** causes **capacity loss** causes **bad performance** and this is a causal relationship.
> >
> > 2. Now, is this thing really **causal** or is it only a **correlation**? The difference is huge, if it's a correlation, then the problem is not that important anymore, because if the agent can recover capacity as soon as it gets new data (or, new data with reward signal), then why do we care about capacity loss? It seems to me when new data come in, capacity loss is automatically solved.
> >
> > Again I'm sorry for the late reply, but if you have time can you please give a short response on
> >
> > 1. to your opinion, is this thing a causation or just correlation? (if it's correlation, then I feel like the paper's story needs to be rewritten a bit)
> > 2. if it's causation, can you please give a short paragraph summarizing what is the strongest evidence in paper that it is causation and not correlation?
> >
> > Another related issue:
> >
> > > This concern seems to stem from different interpretations of “capacity”, and whether it makes sense for a definition of capacity loss to be reversible. Our definition of network capacity is specifically geared towards the network’s ability to perform the kinds of updates that lead to short-term performance improvements. This means a network which has lost capacity hasn’t necessarily lost its ability to ever learn again, but rather that its current parameter values are in an unfriendly region of the optimization landscape from which it is difficult to perform updates that lead to policy improvement, making this definition well-suited to RL.
> >
> > So it seems you are saying capacity is lost, but not entirely lost, and it can recover (though more difficult?) Does that mean your proposed measure of capacity is not really the right measure? If the point is that without INFER, networks are harder to recover, then shouldn't there be a capacity measure that actually **captures how difficult it is to recover**?

---

> > > ### Author Response · Authors · 2021-11-29
> > > **Main Response to Reviewer**
> > >
> > > We thank the reviewer for their detailed explanation of their concerns. This response has highlighted two crucial aspects of the contribution of the paper: how we justify the causal relationship between capacity and performance, and how the two notions of capacity subtly differ in the properties they measure.
> > >
> > > **The Causal Relationship Between Performance and Capacity**
> > >
> > > We first address the reviewer’s concern about causality vs correlation in capacity loss and performance.  We focus on isolating the effect of capacity on performance, independent of other reward signals coming from the environment. We note that the reward signal that the agent observes in the environment will certainly have some effect on performance (i.e. an agent which never sees a reward will never improve its policy, no matter how much capacity it has). Our candidate causal graph therefore has an arrow `Reward Signal` -> `Performance`, and an arrow `Reward Signal` -> `Capacity`. The reviewer’s question is whether we have conclusively shown an arrow from `Capacity` -> `Performance`, or whether the observed correlations can be explained via the reward signal the agent receives.
> > >
> > > We claim that this paper provides strong evidence towards the existence of a causal relationship. This evidence comes in 3 parts: the example of Montezuma’s Revenge in **Figure 3**, where we have **controlled for the exploration algorithm and random seed to isolate the effect of representation collapse on performance**; our study of InFeR in **Figure 5**, which acts as a more direct **intervention on the capacity variable** while preserving both the agent’s exploration strategy/random seed and the value-learning objective, and demonstrates that **increasing network capacity results in performance improvements**; and the **temporal ordering** between capacity and performance improvement revealed in the Sparse Pong example (causes occur before effects, and the effective dimension increase preceded the performance improvement).
> > >
> > > We provide more details to justify these claims in a follow-up comment; if the reviewer is satisfied with the above response, they can continue to our discussion of capacity loss and recovery without reading the additional comment.
> > >
> > > **Loss and Gain of Capacity**
> > >
> > > We measure capacity in two ways: “target-fitting capacity”, and the “effective dimension”. **We only observe an agent regain significant capacity with respect to effective dimension**, not the target-fitting notion of capacity. The reviewer’s concerns therefore appear to stem from the fact that target-fitting capacity, particularly given a sufficiently large training budget, and effective dimension _don’t measure exactly the same thing_.  Effective dimension's tractability comes at a cost: unlike target-fitting capacity, it fails to account for the network's ability to update earlier layers.
> > >
> > > Our notion of target-fitting capacity captures how well an agent can fit new targets given a fixed number of optimization steps. The number of optimization steps used in this definition **determines whether or not we expect to see agents “regain” capacity late in training, and allows the notion of capacity to be tailored to match the timescale relevant to the problem**: for example, using 1M optimization steps is less relevant as a measure of capacity in a sample-efficient benchmark such as Atari 100k. In principle, by giving the optimization algorithm O a sufficient number of steps (e.g. the training budget of the RL problem) in Definition 1, it is possible to construct a version of our target-fitting capacity measure which would capture the network’s ability to “regain” capacity and fit arbitrary function classes. Under such a definition, capacity loss would be permanent. We did not explore this further in the paper due to the computational costs that would be required to measure such a notion.
> > >
> > > Given a small number of training steps, the target-fitting capacity estimator measures whether the network can quickly fit new targets. This is precisely “a capacity measure that actually captures how difficult it is to recover”; a representation that can recover easily from a low effective dimension is precisely one that can fit new targets within relatively few optimization steps. It is further relevant to performance as agents which can quickly update their value functions to fit new targets will be better able to perform corresponding policy improvements and progress more quickly on a given task. In our evaluations, this notion of capacity did not stage dramatic increases late in training, though in some cases our coarse checkpointing approach didn’t detect early-training declines, as can be seen in Figure 7 in Appendix B, which makes the capacity trajectory appear relatively flat.
> > >
> > > We provide additional details to these points in sub-comments.
> > >
> > > [1] Eichler Michael, 2013. Causal inference with multiple time series: principles and problems. Phil. Trans. R. Soc. A.

---

> > > > ### Author Response · Authors · 2021-11-29
> > > > **Sub-response 1: Causal relationship justification**
> > > >
> > > > We provide additional details to flesh out the summary of the evidence for a causal relationship between capacity and performance.
> > > >
> > > > **Controlling for other performance factors**: Our analysis of agents in Montezuma’s Revenge and Sparse Pong in **Figure 3** all use the same epsilon-greedy exploration algorithm and the same set of random seeds. By controlling for the exploration algorithm and random seed, we have significantly limited the variation that can occur in the [Reward Signal] variable. Further limiting this variation by deterministically controlling the sampled transitions from the environment would take us out of the online RL regime. As a result, **since the agents use the same uniform random policy to generate trajectories at the start of training, use the same random seed to initialize their network parameters, and follow the same epsilon-greedy exploration strategy**, we can be reasonably confident that **the difference in performance we see is a result of the properties of the learned representation**.  Importantly, all agents start training with a uniformly random policy, and all follow epsilon-greedy exploration with the same epsilon scheduler. Thus, any differences in their behaviour policy can be attributed to the value function updates they made based on this initial random distribution. This experiment thus isolates the effect of the learning objective on the representation; our next experiment aims to isolate the effect of preserving capacity on performance in a similar way.
> > > >
> > > > **Intervention on capacity:** The cleanest evidence we have for a causal relationship between capacity and performance comes from **intervening on network capacity through the InFeR objective**. The loss we define in InFeR is independent of the environment transition dynamics and reward, allowing us to isolate its effect on the capacity of the learned representation. InFeR significantly improves performance on environments where we tend to see representation collapse, such as Venture and Montezuma’s Revenge, in both Rainbow and DDQN agents. As in our first experiment, agents trained with and without InFeR initially follow uniform random exploration policies, so the effect of InFeR on exploration is uniquely determined by its effect on the agent's value function. As InFeR hasn’t changed the value function loss or the exploration strategy, we deduce that this effect on performance is again due to the learned representation -- in effect, we have attempted to construct a "randomized controlled trial" of performing an intervention on a network's training dynamics that increases its target-fitting capacity. Since the only explicit effect of InFeR on the representation is to preserve its ability to fit a class of target functions, **we conclude that it is the network’s increased capacity that is mediating the performance improvement**.
> > > >
> > > > **Temporal relationship between capacity and performance:** A key tool in identifying causal effects is to study the relationship between two variables over time: if X causes Y, then we expect a change in X to happen before a change in Y [1]. The Sparse Pong agent, where we observed an increase in capacity that preceded the improvement in performance, is therefore actually consistent with the hypothesis that there is a causal relationship between capacity and performance. Had the relationship between capacity and performance been mediated by the environment learning signal (i.e. no arrow [Capacity] -> [Performance]), then we would have no reason to expect the increase in effective dimension to come before the improvement in performance; both could have happened simultaneously, or performance could have increased before capacity. This temporal ordering suggests that the increased effective dimension was necessary for the agent to use the learning signal from the environment for policy improvement. This piece of evidence alone is not sufficient to identify a causal relationship; while the observation is consistent with the hypothesis that representation collapse prevents performance improvement, there could be a hidden variable determining performance which the effective dimension is correlated with. The previous two pieces of evidence provide controlled investigations suggesting that this is not the case.

---

> > > > > ### Comment · Reviewer_Mqcv · 2021-11-30
> > > > > **Response to authors**
> > > > >
> > > > > I want to thank the authors for providing the detailed response and engage in this very interesting discussion. And it seems to me this response has clarified a lot of things. (I feel that in the paper some details are a bit confusing but the response overall makes sense)
> > > > >
> > > > > In the next version of your paper, you might want to consider
> > > > > - incorporate some of the reviewer discussion into your paper (might be good to further emphasize a bit some subtle points in the paper so that your writing is very accurate and your arguments and supportive evidence are very clear to readers), and might be good to have some of this discussion in your appendix (in a re-organized way)
> > > > > - I'm not sure if the "key insight" is currently emphasized in the paper? If not can you discuss it a bit in the paper or put in appendix. And some recent work actually also support your insight, see "Masked Autoencoders Are Scalable Vision Learners" which has ablation showing that the commonly used linear probing/prediction evaluation method in representation learning might not be that good for measuring representation quality. I think this point is very interesting.
> > > > >
> > > > > I thank the authors again for these discussions and your effort and I now increase my score to 6 since I feel a lot of the concerns are indeed addressed.

---

> > > > > > ### Author Response · Authors · 2021-11-30
> > > > > > **Thanks for the fruitful discussion!**
> > > > > >
> > > > > > Thanks to the reviewer for engaging in the discussion period and for their insightful comments. We think these recommendations will greatly improve the paper. We will update the paper to emphasize our justification of the causal relationship between performance and capacity, and to emphasize the distinction between target-fitting capacity and effective dimension, in particular highlighting the limitations of looking only at the last layer of a network. Finally, thanks for the connection to recent work. It looks extremely interesting and we look forward to seeing what insights may be relevant to our paper.

---

> > > > ### Author Response · Authors · 2021-11-29
> > > > **Sub-response 2: effective dimension and target-fitting capacity**
> > > >
> > > > We first provide a few points on the effective dimension:
> > > > 1. It is useful for evaluating whether the agent can quickly update its predictions in response to new reward signals (e.g. by updating weights in its final layer), but doesn’t capture the ability of earlier layers to adapt. This ignorance of early layers is why agents can “regain” effective dimension.
> > > > 2. We use effective dimension in our evaluations because it correlates with the  low-optimization-budget notion of capacity (and so translates well to measuring an agent’s ability to improve its policy), connects to prior work studying network capacity in RL, and requires fewer design choices in its construction.
> > > > 3. However, it does not measure exactly the same thing as the target-fitting capacity measure we use under moderate-to-large training budgets. In our empirical results in the paper, target-fitting capacity (given a sufficient optimization budget) does not exhibit a similar ability to increase late in training as does effective dimension.
> > > >
> > > > This observation highlights a **key insight** of the paper which may prove to be significant for the broader representation learning in RL community: that _considering only the penultimate layer outputs of a network does not provide a full picture of the representation_. Instead, it is useful to consider how the network can perform updates in response to new information from the environment.
> > > >
> > > >     So it seems you are saying capacity is lost, but not entirely lost, and it can recover (though more difficult?) Does that mean your proposed measure of capacity is not really the right measure?
> > > > Ultimately, there is a trade-off between measuring whether a network can _ever_ fit a target function or whether it can fit a target function _in a timeframe relevant for policy improvement_. While effective dimension focuses on the latter, our definition of target-fitting capacity can measure either of these properties depending on its optimization budget. We therefore believe that this is a flexible and powerful framework for understanding network capacity. While we believe lower optimization budgets will be more relevant for capturing the ability of the network to quickly improve its policy, our definition of capacity can also be adapted to consider whether a network will ever be able to represent a given function class.
> > > >
> > > > Finally, we emphasize that in all but one case we saw in our empirical evaluations, the agent was not able to recover from representation collapse. This was despite seeing at least one nonzero reward during training even in extremely sparse settings such as Montezuma’s Revenge. Thus, the answer to the question
> > > >
> > > >     ...if the agent can simply increase capacity when it gets new data with reward, then is this kind of capacity loss really a problem?
> > > > is **yes, because agents cannot always increase capacity when they see a reward**. While we have one example of an agent recovering from representation collapse in Figure 3, we have many more instances in that figure where the reward signal from the environment wasn’t sufficient for recovery on its own -- even in Montezuma's Revenge, the agents all saw at least one non-zero reward at some point during training, but it was only the agents with the non-zero prediction target which were able to translate this reward into policy improvements. Capacity loss and representation collapse are therefore still important phenomena for deep RL practitioners to be aware of.

---

> ### Author Response · Authors · 2021-11-17
> **Response to Reviewer Mqcv (Part 2)**
>
> 3. Clarify the MNIST experiment setup
> We have expanded on our explanation of the MNIST experiments in the main paper to improve the clarity of this section. Concretely, we have added the following text:
> “...We aim to isolate the effect of sequential prediction tasks on capacity loss. To minimize the potential for confounding factors to influence our results, we construct our toy iterative prediction problems on the MNIST data set, which consists of images of handwritten digits and corresponding labels, and manually construct a sequence of targets which the network must fit over the course of training. We first consider labels computed by a randomly initialized neural network $f_\theta$: we transform input-label pairs $(x,y)$ from the canonical MNIST dataset to $(x, f_\theta(x))$, where $f_\theta(x)$ is the network output. To generate a new task, we simply reinitialize the network; our evaluations consist of 10 iterations of label-generation followed by a training period during which we run a gradient-based optimizer on the network starting from the parameters we obtained in the previous iteration. We further consider a `sparse-reward' version of MNIST: for each of 10 iterations $i$, we use the label $\hat{y}_i = 1[y < i]$, where $y$ is the true label of the image. For example, at the first iteration, all images are assigned label zero. At the second iteration, the images of the digit zero are assigned label one, while all other inputs retain the zero label. This continues until all inputs are assigned label one at the final iteration. We follow the same training procedure in both cases: optimizing the network for a fixed number of steps on one set of labels, then generating new labels and running the optimization algorithm again from the parameters obtained in the previous phase. “
> We are happy to answer any additional questions the reviewer may have about this setup.
> 4. Discuss why InFeR sometimes hurts performance:
> InFeR works by regularizing the agent’s representation to preserve random subspaces of its initial value; this may have the side effect of slowing down the optimization process and making policy improvement updates more difficult in environments where many intermediate policies must be evaluated in succession. Additionally, if the network’s initial inductive bias is a poor fit for the environment (for example, if it tends to map states with very different values to similar features), then regularizing towards the initial value of the representation may have the effect of exacerbating this poor inductive bias during training. The latter hypothesis is difficult to evaluate, as it’s not clear exactly how one would measure a “poor inductive bias” universally across environments. However, the former hypothesis is something we look into in Section 4.2: we observe that adding additional degrees of freedom into the final feature representation layer by increasing its width narrows the gap between InFeR and Rainbow in most games where InFeR hurts performance. These results can be found in Figure 7 in the main text and Figure 14 in the Appendix. In other words, InFeR works by limiting the degree to which the network changes its features during training. In some environments, learning features that look very different from their initial values seems  to useful to make fast learning progress. InFeR therefore slows learning progress in these settings, but giving the representation more degrees of freedom to "work around" the subspace regularized by InFeR boosts performance to be competitive with or outperform the unregularized network.
> 5. Updating discussion of data augmentation  in RL:
> The reviewer is correct that there are many existing approaches which have successfully deployed data augmentation in RL. Our point is not that these approaches fail, but rather that translating them from the supervised learning setting often requires some thoughtful design choices. The paper we cite to support this claim, for example, notes that in order to obtain unbiased policy gradient updates, it is crucial to apply the same transformation to both the input and target state. Doing otherwise can introduce bias into the optimization process in a way that doesn’t occur in supervised learning. We will update the paper text to better reflect this.

---

### Official Review · Reviewer_zrcc · 2021-11-02

**Correctness:** 4
**Technical Novelty And Significance:** 2
**Empirical Novelty And Significance:** 3
**Recommendation:** 8
**Confidence:** 5

**Details Of Ethics Concerns:**

Ethics Concerns:
None

**Main Review:**

Strengths:
- Very nice paper that is well corroborated and is clearly presented in logical order
- Solid experiments, good job!
- Great analysis in Section 4.2

Weaknesses:
- Needs to better logically connect 3.1 and 3.2.  Why is effective dimension representative of/correlated to capacity? A good mathematical proof would work, ideally in Appendix B.
- INFER seems very similar to RND exploration, except that the L2 error is moved from intrinsic reward to the loss function. Ablate on this! (such as Rainbow + RND)
- Novelty of problem. This problem has also been shown in supervised learning, where neural networks become less adaptable over time, and I believe in several other works in RL, such as in Kumar's prior work. The paper, especially in introduction, present it like it is a new problem. Good to downplay this.
- Figure 3 and Figure 4 make the same message (correlation between effective dimension and performance). Recommend combining both figures and move some repeated information (such as 4(b)) to the Appendix.
-In Appendix, show how robust INFER is against hyperparameter tuning. It looks like beta=100 is a pretty arbitrary value.
-Paper seems to be rooted in Atari environments. What prevents extending this to continuous control environments such as Mujoco?

Questions:
- In Figure 2 (MNIST experiments) (clarification), is the label y_i = 1[y<i] a mask applied over the random network output?

**Summary Of The Paper:**

Summary:
- Over course of training, deep RL agents experience "capacity loss", where networks are unable to quickly fit new functions. This problem is further exacerbated by non-stationary predictions (such as bootstrapping) over the course of training.
- To prove this problem, the author runs two experiments (Atari and MNIST), and show that training error against a randomly-initialized network increases over time. They also define effective dimension (rank of feature space) and show that effective dimension is tightly correlated with performance.
- The authors introduce INFER, which regularizes Q-value loss with error w.r. to randomly initialized network, and evaluate this on Atatri-57, showing most benefits in sparse reward environments (Montezuma's revenge)

**Summary Of The Review:**

Overall:
This is a well written paper which gives strong evidence that capacity loss/representational collapse is indeed a root problem for deep RL. As long as the authors address the weaknesses, I am willing to bump this up from weak accept to strong accept.

---

> ### Author Response · Authors · 2021-11-17
> **Response to Reviewer zrcc (Part 1)**
>
> We thank the reviewer for their thoughtful and insightful review. We appreciate their agreement on the novelty and significance of our results, along with the concrete recommendations to communicate the ideas presented in the paper more clearly and rigorously. We include responses to specific questions and comments below.
>
> 1. **Clarifying the connection between effective dimension and capacity.**
> The relationship between effective dimension and target-fitting capacity is nuanced. At the extreme, agents suffering from representation collapse tend to have both low effective dimension and low target-fitting capacity. This is best exemplified by QR-DQN agents on Montezuma’s Revenge. However, effective dimension is likely to measure a number of other properties of the agent’s representation and of the data-generating distribution induced by the agent’s policy interacting with the environment which act as confounders. For example, agents which visit a more visually diverse distribution of states might obtain a higher effective dimension as computed by our estimator than counterparts with identical representations. It is also not obvious that a large effective dimension is necessarily beneficial once the representation has “sufficient” capacity: a tabular  representation would have a high effective dimension but would have, for example, poor generalization properties and so might not be preferred over a representation with lower effective dimension but could better generalize to new states. We therefore recommend interpreting the effective dimension as a noisy signal of representation capacity, one which accurately detects extreme forms of capacity loss but can suffer from some confounding factors. Critically, we note that in Figure 4, the data points for the challenging Atari games tend to cluster along axes: the games with “full rank” tend to attain non-zero performance, while the games which attain 0% human-normalized score tend to be those with lower effective dimensions. In between these two extremes, however, there isn’t a straightforward linear correlation. We think understanding these additional factors affecting effective dimension, in particular the effect of state similarity and generalization between states, presents an exciting direction for further research.
> It is not straightforward to provide a proof of a fully generic result relating target-fitting capacity and effective dimension due to the complexity of neural network training dynamics. However, in our discussion of **Corollary 1** in the newly-added section of Appendix A, we do provide some intuition for why the two measures will be in agreement when the representation has collapsed. Specifically: when the representation has collapsed, the effective dimension will trivially be zero, and if this collapse occurred under stochastic optimization in the presence of ReLU units it is likely that all units will be saturated, preventing the network from fitting future targets. We are currently exploring whether it is possible to make this intuition more rigorous beyond the result currently in the Appendix.
> 2. **Robustness of InFeR to hyperparameters.**
> We have begun a sweep over hyperparameter values and will update the appendix to include these results once the experiments have finished running.
> 3. **Similarity to and distinction from RND.**
> We note that the distillation objective in our context is not used as an exploration signal. Conversely, the RND loss is not directly used as a representation-learning method, but rather as a means of providing exploration bonuses. The focus of our paper is on isolating the effect of agents’ representation capacity on their ability to make learning progress. However, we note that the RC-DQN agent is following an objective similar to the RND loss by fitting a value function to auxiliary rewards generated by a randomly initialized network. In this case, the RC-DQN loss seems to interfere with the main task in dense-reward games, resulting in poor performance in e.g. the Seaquest environment. The loss does prevent representation collapse in Montezuma's Revenge, though we note it does not outperform InFeR.
> 4. **Redundancy between Figure 3 & 4**
> This is a useful observation, and we will work on improving the efficiency of our visualizations to provide additional space for discussion of some of the nuances brought up in the paper’s reviews. We will provide an update to the forum when these changes are reflected in the pdf.

---

> > ### Author Response · Authors · 2021-11-17
> > **Response to Reviewer zrcc (Part 2)**
> >
> > 5. **MNIST experiment clarifications**
> > We have included more details of the MNIST experiment setup in the main body of the paper. The paragraph now reads as follows:
> > “...We aim to isolate the effect of sequential prediction tasks on capacity loss. To minimize the potential for confounding factors to influence our results, we construct our toy iterative prediction problems on the MNIST data set, which consists of images of handwritten digits and corresponding labels, and manually construct a sequence of targets which the network must fit over the course of training. We first consider labels computed by a randomly initialized neural network $f_\theta$: we transform input-label pairs $(x,y)$ from the canonical MNIST dataset to $(x, f_\theta(x))$, where $f_\theta(x)$ is the network output. To generate a new task, we simply reinitialize the network; our evaluations consist of 10 iterations of label-generation followed by a training period during which we run a gradient-based optimizer on the network starting from the parameters we obtained in the previous iteration. We further consider a non-stationary binary classification version of MNIST: for each of 10 iterations $i$, we use the label $\hat{y}_i =1[y < i]$, where $y$ is the true label of the image. For example, at the first iteration, all images are assigned label zero (similar to predicting the value of a poorly-performing policy in a sparse-reward environment). At the second iteration, the images of the digit zero are assigned label one, while all other inputs retain the zero label. This continues until all inputs are assigned label one at the final iteration. We follow the same training procedure in both cases: optimizing the network for a fixed number of steps on one set of labels, then generating new labels and running the optimization algorithm again from the parameters obtained in the previous phase. "
> > We will be happy to answer any additional questions the reviewer may have about this setup.

---

> > > ### Comment · Reviewer_zrcc · 2021-11-29
> > > **Response to Rebuttal**
> > >
> > > Thanks for the detailed reply! The authors have responded to all of my concerns and have/will address them in the Appendix.

---

### Official Review · Reviewer_KLqH · 2021-11-05

**Correctness:** 4
**Technical Novelty And Significance:** 4
**Empirical Novelty And Significance:** 4
**Recommendation:** 8
**Confidence:** 4

**Main Review:**

This is a very well written and very insightful work. I took great pleasure and interest reading it.
The writing is flawless, the structure is very clear and the presentation of ideas unfolds naturally.
The contributions are significant, their evaluation is convincing, both in the method and in the results, and the discussion has an appreciable depth.

I identify two weaknesses. First, some evaluation aspects are omitted, and I question their effect on the mechanism of InFeR. Second, some parts of the discussion could maybe be condensed to develop other aspects more in-depth, both in the main text or as references to the appendix. The following remarks list the aspects upon which I would like to see more details and/or an improved discussion in the paper.

The introduction of network capacity and, more importantly, the characterization of a network's effective dimension are well-developed ideas, with enough nuance with other approaches from the literature to be relevant contributions in themselves. In a sense, the effective dimension seems very similar in spirit to a VC dimension. Could the authors elaborate a little on this (both here and in the paper)?

In Figure 3, I fail to understand why the initial effective dimension of the QR-DQN network is always the smallest, except for the sparse pong task. This seems very counter-intuitive and should be justified.

The idea to regularize against a set of projections of the initial features is equivalent to defining a set of corresponding auxiliary tasks. I would appreciate seeing more details on how these $g_i$ projections are designed in the first place. Are they random projections? Is there a rationale in their construction? How dependent on their choice is the overall training?

One common remark on Deep RL papers is the very low number of seeds used to assess the statistical significance of experiments. This paper makes no exception: the canonical number of 3 seeds is used. I am well aware of the compute cost issue and I don't see this number of seeds (despite the associated low statistical confidence) as a reason to cross out these results. Nevertheless, the presented experiments display large variance in the results. Specifically, the performance gain of InFeR seems to come along an strong increase in performance variance. It seems unavoidable to discuss the causes for this variance. Is there a link with the construction of $g_i$? Or the variance of gradients for the ADP loss in sparse rewards environments? Is it related to the games InFeR is tested on? What do the corresponding policies actually do (a video would definitely be nice)? Do they act well consistently? I believe the discussion could be quite deeper on this question.

In the testing of hypothesis 1, concatenating with a fully random network is a rather radical comparison point. First, I believe the way this network is initialized might have an impact, and this should be at least mentioned explicitly (if not discussed). Second, I believe an in-between solution would consist in adding lateral connections, in a fashion similar to that of progressive neural networks. In a sense, this can be compared to adding random projections after each layer. This might provide a few more insights regarding the claim that the effect of InFeR on earlier layers is crucial to its success.

An interesting aspect which I would have liked to see is the evolution of the effective dimension at different depths of the network. Does representation collapse occur abruptly at a certain depth? Is it caused by early or deep layers in the network? Does it manifest within convolutional layers? Do shared weights (under-parameterization) act as a regularizer? By the way, what is the network architecture used? Classical DQN-style 3 convolutional layers + 2 fully connected ones (with the appropriate variations for QR-DQN)? What would happen with deeper networks such as the IMPALA architecture?

Minor remark:
Page 2. I believe the $(X_t, A_t, R_t, X_{t+1})$ tuple of equation 2 was actually introduced as $(x_t, a_t, r_t, w_{t+1})$ a few lines above. Making this consistent would help.

**Summary Of The Paper:**

This paper proposes a new regularization method for deep reinforcement learning, that is designed to prevent representation collapse and thus retain the ability of the network to fit functions that evolve in time, such as the sequence of Q-functions generated by approximate dynamic programming methods. This method consists in regularizing the network's features towards a set of initial linear transformations of their (the features) values at initialization.

**Summary Of The Review:**

Overall this is a solid contribution on the question of preventing capacity loss in deep RL. The ideas are novel and the point of view developed on representation collapse, and the proposed way to prevent it, is a significant contribution to the community. A number of points could fruitfully be discussed, both in the main text and in the appendix, which would increase the practical impact of the paper.

---

> ### Author Response · Authors · 2021-11-17
> **Response to Reviewer KLqH**
>
> We thank the reviewer for their thoughtful and positive feedback. The reviewer has highlighted the significance and novelty of our observations, while also identifying areas where we can improve the exposition of the core ideas of the paper. We have endeavoured to address remaining concerns in our response.
>
> 1. a) **Concerning statistical significance**
> We agree that statistical significance of results in deep RL is crucial for drawing robust and generalizable conclusions, especially in empirical works such as this one.
> The paper has two principal empirical claims that can be considered: first, that deep neural networks used in RL tasks can lose their capacity to fit target functions over the course of training; and second, that InFeR mitigates this phenomenon to some degree and improves performance. The first claim is challenging to rigorously show in deep RL environments due to the many moving parts of these algorithms, and we will discuss this further in the next bullet point. The second claim, however, is computed by taking a median over 57 different environments, resulting in 171 (non-iid) evaluation samples for each method.
> We therefore present the claim that InFeR improves the median performance of Rainbow over the Atari 2600 environments with reasonably high confidence. Within individual games it is not always the case that the effect of InFeR on performance is clear. In some environments such as Montezuma’s Revenge, where the empirical distribution of evaluation returns for each method do not overlap, we can straightforwardly conclude the direction of the effect of InFeR, but for other environments such as Seaquest, the effect size is well within the standard error. Nonetheless, because our main claims refer either to specific classes of environments where effects are more pronounced, or to global properties such as the effect of InFeR on median performance over all 57 environments, ambiguous statistical results on single environments are not of dire concern for the main interests of the paper. To be thorough we are currently training additional runs to gain greater confidence in the outcomes of our per-game results.
> Part of the reason why it is so difficult to obtain statistical significance of claims in RL is because of the large variance in performance of many methods over different random seeds or small hyperparameter tweaks. Further, the many moving parts in popular deep RL algorithms mean that it is often difficult to isolate the effect of a single variable and draw conclusive observations. We therefore highlight our study of the non-stationary MNIST environment as a means of isolating this phenomenon in a rigorous way while reducing the number of samples necessary to identify effects. While we only run a handful of seeds on each task, we again see a sufficiently strong effect supporting a) that the non-stationary prediction tasks reduce network capacity over time and b) that InFeR reduces this effect that we do not believe additional seeds would provide significant information gain.
>
>      b)  **A  note on effect sizes.** Our study of capacity loss in Atari environments runs into two key subtleties: first, our study of the MNIST setting reveals that capacity loss is most pronounced when the network is underparameterized relative to its prediction task. It is likely that the difficulty of the nonstationary prediction problem in each game varies, and therefore the shared network architecture may be relatively over- or under-parameterized depending on the evaluation task. As a result, we do not expect to see a consistent effect size across the Atari environments; indeed, for a sufficiently overparameterized model we might even expect to see some positive transfer as the network becomes better at distinguishing visually similar inputs which correspond to states with different values.
>
> 2. **Questions re: Hypothesis 1**: we apologize for the lack of clarity in the description of our evaluation of Hypothesis 1. To clarify: we use the same network initialization for the concatenated feature. Effectively, this is equivalent to preserving a single (axis-aligned) dimension of the features the network had at initialization, while not influencing the gradients propagated back to the earlier layers of the network except through its effect on the loss. The version of InFeR used in Figure 6 also uses a single auxiliary head, and so this allows us to disentangle the effect of giving the value prediction head access to a low-dimensional subspace of randomly-initialized features  from the effect of the objective on the whole-network training dynamics.
>
>
> (Continued in next comment)

---

> ### Author Response · Authors · 2021-11-17
> **Response to Reviewer KLqH (Part 2)**
>
> 3. **VC dimension connection.**
> This is a nice observation: our notion of target-fitting capacity bears some resemblance to the VC dimension by measuring the network’s ability to arbitrarily group together different inputs, though we use a much fuzzier notion (average error on a class of functions on a fixed distribution of inputs rather than maximal size of inputs which can be shattered) which is easier to measure. Our target-fitting capacity measure is also distribution-dependent, which means it is easier to tailor to the types of inputs an RL agent will encounter in a given environment, and the types of targets we want it to be able to fit.
> more details on random head construction
> 4. **Construction of random projections:** the random heads used in the InFeR task are generated by the same network initialization scheme used to instantiate weights for the rest of the network; we do not explicitly set them to be orthogonal at initialization, though it would be interesting to see whether this would have any effect on the resulting learning dynamics.
> 5. "...I would have liked to see is the evolution of the effective dimension at different depths of the network."
> This is a really interesting idea: we chose to focus on the final layer because the effective dimension is then more straightforwardly measuring how well the final set of weights can linearly distinguish input states, but there is no theoretical barrier to also measuring the effective dimension at any layer of the network. Given prior work studying e.g. information bottlenecks, we might expect greater state compression at the final layer compared to earlier-layer features, resulting in higher effective dimensions for earlier layers. This analysis might also enable us to pinpoint more clearly the mechanisms by which representation collapse occurs in neural networks trained on sparse-reward environments. We will look into this further and endeavour to include some of our analysis in Appendix B.

---

### Author Response · Authors · 2021-11-17
**Thanks to all reviewers**

We’d like to thank all reviewers for providing thoughtful, insightful, and actionable reviews. We address each reviewer’s individual concerns in separate responses, and summarize our current and planned updates to the paper here. We note that some of the additional evaluations have not completed running at this time, and we will make a note of this where relevant and post an update in the discussion thread when these results have been added to the pdf.

1. We have **updated the description of the MNIST experiment** so that it provides a more detailed account of how the task is constructed.
2. We have more **explicitly written out the theoretical motivation** of our study, including citations of prior works showing that under certain assumptions TD learning dynamics provably lead to feature collapse in idealized sparse-reward settings. This description also provides intuition behind the lower effective dimension of the QR-DQN agents in Figure 3. Full details are now in **Appendix A** of the revised submission.
3. We have **revised our presentation of the results in Section 3.2** to give a more nuanced comparison between the network properties measured by effective dimension and target-fitting capacity. A summary of this is included below.
4. We will include a more detailed empirical analysis of the trajectories followed by a representative set of agents on the first 10M frames in the Atari environments analyzed in this paper, resolving some of the queries raised regarding the MZ experimental results in Figure 4(b). The current versions of these analyses are included in Appendix B, and additional agents and sources of random predictions are currently under evaluation.
5. We plan to include a number of additional empirical evaluations in the appendix, including a hyperparameter sweep and additional seeds for some of our empirical evaluations. These evaluations are still underway, but we will post a note when they are represented in the paper.

We thank the reviewers for their recommendations, and think that the revisions described above have significantly strengthened the paper. In particular, R3 and R4’s comments have allowed us to significantly improve the clarity of the messaging of the paper, to emphasize the following nuances, which we summarize here.

1. We do not claim that all RL agents lose capacity over the course of training in all environments, nor do we claim that when this capacity loss occurs it necessarily hinders performance.
    - We claim that in settings where the neural network function approximator is exposed to a series of complex (relative to the size of the network) and highly variable prediction tasks, capacity loss often occurs, and in settings where the architecture is already somewhat under-parameterized for the task at hand it appears to slow down learning progress [Figures 1 and 2].
    - We have updated Sections 1 and 3 of the paper to make this clearer to the reader, softening the statement that “RL agents lose capacity over the course of training” to more accurately reflect the fact that while capacity loss is evident in many environments, there do exist some tasks where it is difficult to measure.
2. We do not claim that effective dimension and target-fitting capacity are both measuring exactly the same thing.
    - Instead, we claim that target-fitting capacity and effective dimension both measure _similar but non-identical notions of capacity_. The two measures converge for networks whose representations have collapsed and/or whose activation units have all saturated, in which case both effective dimension and target-fitting capacity will be low. However, it is possible for a network to have low effective dimension yet high target-fitting capacity (for example, if the features have small norm, or if the layer outputs are dominated by a bias term), or vice versa (for example, uniformly at random mapping inputs to one of $d$ one-hot feature vectors). The two are correlated in that **a representation which can effectively linearly distinguish between input states will generally be able to more effectively fit arbitrary target functions** than one which maps all states to the same linear subspace.
    - Our previous version of the paper hinted at this distinction and provided some thoughts the types of environments where one might be more informative of performance than the other, and so we have refined and emphasized the sections where this discussion occurs in the paper to make sure that this is clear to the reader.

---

> ### Author Response · Authors · 2021-11-24
> **Final update to PDF**
>
> Thanks again to all reviewers for their careful evaluation of our work. We have updated our submission to include additional results promised in bullet points 4 and 5 of the parent comment. Specifically:
> 1. We include a more detailed empirical analysis of the first 10M frames of training before the first checkpoint is sampled in our previous evaluation regime, with results shown in Figure 7, Appendix B. We show that capacity loss does indeed occur in Montezuma’s Revenge, and that agents’ representations experience significant changes in this early regime before stabilizing as is seen in the figures in the main paper.
> 2. We additionally provide contour maps showing the robustness of our method to the choice of hyperparameters, see Figures 8 and 9 in Appendix B.
>
> We look forward to engaging in the remainder of the discussion period to resolve any remaining questions or conceptual concerns.

---

### Decision · Program_Chairs · 2022-01-20

**Decision:**

Accept (Spotlight)

**Comment:**

The paper analyzes the learning behavior of deep networks inside RL algorithms, and proposes an interesting hypothesis: that many of the observed difficulties in deep RL methods stem from _capacity loss_ of the trained network (that is, the network loses the ability to adapt quickly to fit new functions). As the paper points out, some of these difficulties have popularly been attributed to other causes (such as difficulties in exploration) or to less-specific causes (such as reward sparsity: the paper proposes that capacity loss mediates observed problems due to sparsity).

The paper investigates its hypothesis two ways: first by attempting to measure how capacity varies over time during training of existing deep RL methods, and second by proposing a new regularizer to attempt to preserve capacity. These experiments are set up well, and their results are convincing &mdash; while there is likely no perfect way to measure or preserve capacity, the methods chosen here make sense.

This is a strong paper: it proposes a creative, appealing, and interesting hypothesis about an important problem (difficulties in training deep RL methods), and conducts a well-designed evaluation. We expect and hope that it will inspire interesting follow-on work.

We thank the authors for their thorough and helpful participation in the discussion period, including updates to improve the clarity of the paper.